

# Simulation of sub-millimetre atmospheric spectra for characterizing potential ground-based remote sensing observations

Emma C. Turner[1,a], Stafford Withington[1], David A. Newnham[2], Peter Wadhams[3], Anna E. Jones[2], and Robin Clancy[3]

[1]Cavendish Laboratory, University of Cambridge, Madingley Road, Cambridge, CB3 0HE, UK
[2]British Antarctic Survey, High Cross, Madingley Road, Cambridge, CB3 0ET, UK
[3]Department of Applied Mathematical and Theoretical Physics, University of Cambridge, Wilberforce Road, Cambridge, CB3 0WA, UK
[a]now at: Met Office, FitzRoy Road, Exeter EX1 3PB, UK

*Correspondence to:* Emma Turner (emma.turner@metoffice.gov.uk)

**Abstract.** The sub-millimetre is an understudied region of the Earth's atmospheric electromagnetic spectrum. Prior technological gaps and relatively high opacity due to the prevalence of rotational water vapour lines at these wavelengths have slowed progress from a ground-based remote sensing perspective; however, emerging superconducting detector technologies in the fields of astronomy offer the potential to address key atmospheric science challenges with new instrumental methods. A site study, with a focus on the polar regions, is performed to assess theoretical feasibility by simulating the downwelling clear-sky sub-millimetre spectrum from 30 mm (10 GHz) to 150 $\mu$m (2000 GHz) at six locations under annual mean, summer, winter, daytime, nighttime and low humidity conditions. Vertical profiles of temperature, pressure and 28 atmospheric gases are constructed by combining radiosonde, meteorological reanalysis, and atmospheric chemistry model data. The sensitivity of the simulated spectra to the choice of water vapour continuum model and spectroscopic line database is explored. For the atmospheric trace species hypobromous acid (HOBr), hydrogen bromide (HBr), perhydroxyl radical (HO$_2$) and nitrous oxide (N$_2$O) the emission lines producing the largest change in brightness temperature are identified. Signal strengths, centre frequencies, bandwidths, estimated minimum integration times and maximum receiver noise temperatures are determined for all cases. HOBr, HBr and HO$_2$ produce brightness temperature peaks in the mK to $\mu$K range, whereas the N$_2$O peaks are in the K range. The optimal sub-millimetre remote sensing lines for the four species are shown to vary significantly between location and scenario, strengthening the case for future hyperspectral instruments that measure over a broad wavelength range. The techniques presented here provide a framework that can be applied to additional species of interest and taken forward to simulate retrievals and guide the design of future sub-millimetre instruments.

## 1 Introduction

Atmospheric emission at sub-millimetre wavelengths which constitutes part of the longwave energy budget is rich in information on water vapour, trace gases and ice clouds. First reviewed by Harries (1977), the author notes in his complementary paper on the potential for sub-millimetre radiometry that atmospheric measurement in this region has received relatively little



attention compared with other bands such as the infrared (Harries, 1980). After nearly four decades this is still the case. At wavelengths shorter than one millimetre atmospheric absorption is dominated by the pure rotational transitions of water vapour, which can obscure the signals of other constituents from a ground-based vantage point. The wider availability of transmissive atmospheric windows at both longer and shorter wavelengths has therefore put the sub-millimetre region at a disadvantage.

Additionally, a gap in the technology between the electronic and optical devices developed for the microwave and infrared, respectively, the so-called 'terahertz gap' (Sirtori, 2002), has slowed progress. However, recent developments in astronomy technology, particularly in the field of superconductivity - see the SPIE conference proceedings series on *Millimeter, Submillimeter, and Far-Infrared Detectors and Instrumentation for Astronomy (I-VII)* (Holland and Zmuidzinas, 2014, 2012; Arnold et al., 2010; Duncan et al., 2008; Woody et al., 2004; Phillips and Zmuidzinas, 2003) - are providing new opportunities to open

this spectral region up to the atmospheric science community.

The astronomical community has been motivated to overcome these difficulties as sub-millimetre sources from outer space contain critical information about the early universe (Phillips, 1988). Several large-scale observing systems have been built to house instruments that measure at these wavelengths (see Table 1 of Schneider et al. (2009)). For example, the Atacama Large Millimetre/sub-millimetre Array (ALMA) located in the Atacama Desert, Chile which became operational in 2011 comprises

66 radio-telescopes and is the largest astronomical project in existence (Wootten et al., 2009). Using existing radio-telescope sites combined with emerging technology offers a novel and elegant way for the atmospheric sciences to gain insights into an underexploited wavelength range.

This paper presents the requirements for measuring atmospheric species from the ground using their sub-millimetre spectra. The aims of this work are to develop a methodology for simulating clear-sky sub-millimetre (10-2000 GHz) atmospheric spectra

and determining optimal spectral lines and instrument characteristics for ground-based remote sensing of atmospheric trace species. We also identify complementary areas of improvement for spectroscopic reference data. A strategy is established for theoretically characterising the principle measurement requirements for a sample of compounds at selected sites using forward radiative transfer simulations. We focus on polar locations, as these are regions that are particularly vulnerable to climate change (Marshall et al., 2014; Serreze and Barry, 2011), and have unique atmospheric conditions that present a challenge for

instrument deployment.

Four molecules; hydrogen bromide (HBr), hypobromous acid (HOBr), the hydroperoxyl radical ($HO_2$), and nitrous oxide ($N_2O$), are used as example species, chosen because they cover a range of scientific applications, geographical distributions and signal intensities. They are climatically important for the following reasons. HBr and HOBr are part of the bromine cycle, which is of interest because active forms of bromine (BrO and Br) have been shown to strongly catalyse ozone depletion

(McConnell et al., 1992). As well as anthropogenic bromine compounds (which contribute to stratospheric ozone destruction), there are natural sources of bromine which destroy ozone in the troposphere. At high latitudes these sources are associated with the polar sea ice zone, but the precise sources and mechanisms of release are not yet clear (Abbatt et al., 2012; Simpson et al., 2007). HBr is the main bromine reservoir, thus providing a means of removing Br from the atmosphere through wet and dry deposition in the lower troposphere. Continuous measurements of HBr, particularly in the polar troposphere, would

help constrain the distribution and origins of bromine. Similarly HOBr is an important intermediary product in the ozone





destruction process, which is photolysed to produce Br and hydroxyl radicals (OH) without reforming the original ozone it was constructed with. Measuring atmospheric HOBr is particularly challenging because of its low concentration and short lifetime (~10 minutes), and it has only recently been measured in isolation (Liao et al., 2012). Both bromine species have weak spectral signals due to their low concentrations, however, in the case of HOBr, no spectral transitions have been measured yet

above 9,440 GHz. $HO_2$ is also part of the bromine and chlorine cycles, where it is a precursor for HBr and HOBr amongst other reactions. It is a member of the $HO_x$ chemical family ($HO_x = HO_2 + OH + H$) which catalyse ozone destruction in the upper stratosphere and mesosphere (Clancy et al., 1994). Work to quantify $HO_2$'s role in this process is ongoing (Millán et al., 2014). $N_2O$ is a greenhouse gas that is estimated to produce the third largest radiative forcing (excluding water vapour) of all anthropogenic gases behind carbon dioxide and methane (Pachauri et al., 2014). It is an extremely important ozone-depleting

substance, and is expected to remain so throughout the 21st century (Ravishankara et al., 2009). Because of its long lifetime it is well mixed throughout the troposphere and has a significantly stronger spectral signal than the other three target gases in this study.

Like most regions in the electromagnetic spectrum, sub-millimetre wavelengths, alternatively known as terahertz frequencies, have arbitrarily-defined boundaries and depending on those chosen can overlap with parts of the far-infrared and, despite

its name, the millimetre/microwave. For the present study we define the sub-millimetre as 30 mm (10 GHz) to 150 $\mu$m (2000 GHz) incorporating some of the longer wavelengths, which is beneficial when considering emerging technologies in this area. For the remainder of this publication units of frequency are used.

The rest of this paper is organised as follows. A review of atmospheric sub-millimetre observations performed to date is presented in Section 2. Section 3 describes the method used to construct input climatologies, simulate radiances and deter-

mine measurement characteristics. Resulting background simulations over a broad frequency range are presented in Section 4. Signals for the four sample gases are shown in Section 4.2 and each one discussed in turn. Finally, all signal parameters and measurement requirements are listed for each species in Tables 5, 6, 7 and 8.

## 2   Sub-millimetre atmospheric observations

The concept of combining astronomy with atmospheric observations has been demonstrated by the Sub-millimetre and Mil-

limetre Radiometer (SMR) (Frisk et al., 2003), a primary payload of the Odin satellite (Nordh et al., 2003) launched in 2001. SMR has four tunable heterodyne receivers that cover frequencies between 486-504 GHz and 541-581 GHz, and a fifth fixed channel at 118.75 GHz. It shares observation time between astronomy mode and terrestrial mode and has been instrumental in observing atmospheric constituents such as ClO, $N_2O$, $HNO_3$, $O_3$, water vapour and ice clouds (Urban et al., 2005; Ekström et al., 2007). A similar approach was employed by the Microwave Instrument for the Rosetta Orbiter (MIRO) on the Rosetta

spacecraft which was temporarily diverted from its primary purpose, to rendezvous with the 67P/Churyumov-Gerasimenko comet, in order to measure the terrestrial atmosphere during a scheduled Earth flyby (Jiménez et al., 2013). Data recorded in MIRO's channels centred at 183 GHz, 190 GHz, 557 GHz and 562 GHz were used to test the instruments performance against expected results from radiative transfer models, and measurements from the Earth Observing System Microwave Limb





Sounder (EOS MLS). EOS MLS is a sub-millimetre instrument dedicated to terrestrial observations which has operated on the Aura satellite since 2004 (Waters et al., 2006). It has five heterodyne radiometers centred on 118 GHz, 190 GHz, 240 GHz, 640 GHz and 2.5 THz, which have been used to obtain profiles of stratospheric and mesospheric trace gases including HCl, HOCl, $N_2O$, OH, CO and volcanic $SO_2$ (Froidevaux et al., 2006). The sub-millimetre atmospheric spectrum was also

investigated during seven months of observations made by the Superconducting Submillimeter Wave Limb Emission Sounder (SMILES), deployed on the International Space Station. SMILES recorded terrestrial radiation in three bands covering 624.32 - 625.52 GHz, 625.12 - 626.32 GHz, and 649.12 - 650.32 GHz using superconductor-insulator-superconductor (SIS) mixers for sensitive measurements from which vertical profiles of HCl, ClO, $HO_2$, BrO, $HNO_3$ and isotopes of $O_3$ have been retrieved (Kikuchi et al., 2010).

The sub-millimetre part of the spectrum has unique value in the passive remote sensing of cirrus clouds owing to the comparable size of ice particles and wavelengths between 300 - 1000 GHz, leading to enhanced attenuation and possible detection (Prigent et al., 2006; Yang et al., 2003). There is a pressing need to characterise ice clouds as they exert a strong influence on the radiative energy balance yet many uncertainties associated with their microphysical properties remain, particularly those surrounding global distributions of ice water path and particle size (Buehler et al., 2012). Prototype instruments developed

to measure ice cloud properties in the sub-millimetre include the Far Infrared Sensor for Cirrus (FIRSC) (Vanek et al., 2001; Evans et al., 1999), the Compact Instrument for Longwave Cirrus Observations (CILCO) (Hayton and Ade, 2007) and the International Sub-Millimetre Airborne Radiometer (ISMAR) (Charlton et al., 2009). The new generation of ESA/EUMETSAT Earth Observation satellites, Meteorological Operational Satellite - Second Generation (MetOp-SG), scheduled for launch in 2022, will carry the Ice Cloud Imager (ICI) which will measure cloud ice water path in the 183 - 664 GHz range (Thomas et al.,

2012). Additionally, two microwave instruments will fly alongside ICI; MicroWave Sounder (MWS) and MicroWave Imager (MWI), all three of which benefit from state-of-the-art Schottky diode based mixers (Thomas et al., 2014). The analysis in the remainder of this study is restricted to clear-sky atmospheric applications of the sub-millimetre.

While satellite and spacecraft observations have the advantage of large-scale coverage and a favourable vantage point to detect and profile the vertical distributions of high altitude gases they are limited in terms of their payload, and once in orbit

cannot be easily adjusted. Setting up a mission is also time and financially costly, typically spanning decades between concept and launch, by which time technology has moved on. Instruments designed to operate from the ground can offer greater flexibility and are also substantially less expensive to build, deploy, and operate. Whereas polar orbiting satellites will pass over any one high latitude point a maximum of twice a day, Earth-bound instruments benefit from continuous measurements enabling the local diurnal variability of atmospheric parameters to be studied. Although no ground-based instrument has been

built by the atmospheric community to measure the full sub-millimetre spectrum to date, this has been achieved by Fourier transform spectrometers (FTS) which are used as a calibration tool for astronomical telescopes (Matsushita and Matsuo, 2003; Pardo et al., 2004). For example, the Smithsonian Astronomical Observatory (SAO) sub-mm FTS measures continuously from 300 - 3500 GHz at a 3 GHz apodized resolution (Paine et al., 2000; Paine and Turner, 2013) and has been employed by Turner et al. (2012) for studying the terrestrial atmosphere above the Atacama Desert. Additionally, Pardo et al. (2002) used the

Caltech Submillimeter Observatory (CSO) FTS (Pardo et al., 2001b; Serabyn et al., 1998), over the 350 - 1100 GHz range to




constrain estimates of the water vapour continuum and to refine the Atmospheric Transmission at Microwaves (ATM) radiative transfer model (Pardo et al., 2001a). Other such instruments, for example the South Pole FTS (Chamberlin et al., 2003), could also potentially be used in this manner.

Accurate quantification of the different components of the Earth's energy balance measured at the surface and top of at-
mosphere (TOA) is of vital importance for monitoring climate change (Stephens and L'Ecuyer, 2015; Stephens et al., 2012). Improvements are the result of using precise, well-calibrated measuring equipment in a wide range of locations with increased spectral coverage and finer sampling resolution. There has been increased interest within the climate sciences in the far-infrared spectrum, typically taken as 2000 - 20000 GHz (15 - 150 $\mu$m), e.g. see Harries et al. (2008); Turner et al. (2015). This has also been sparsely measured in the atmosphere for similar reasons to those of the sub-millimetre, despite representing up to
half of the total outgoing longwave radiation and three quarters of the total incoming longwave radiation. Recent instruments designed to fill this gap include the Atmospheric Emitted Radiance Interferometer (AERI) (Turner et al., 2004), the Radiation Explorer in the Far InfraRed (REFIR) (Esposito et al., 2007; Palchetti et al., 2008), the Tropospheric Airborne Fourier Transform Spectrometer (TAFTS) Green et al. (2012), the Interferometer for Basic Observation of Emitted Spectral Radiance of the Troposphere (I-BEST) Masiello et al. (2012) and the Far-Infrared Spectroscopy of the Troposphere (FIRST) instrument
(Mlynczak et al., 2004). Whereas all of these instruments were designed to make consecutive measurements covering a wide spectral range, existing sub-millimetre instruments typically measure in narrow channels (excepting the SAO sub-mm FTS), in line with their primary purpose of detecting pre-selected lines of specific gases. However, for the purpose of studying the sub-millimetre's contribution to the Earth's energy budget broadband measurements are required.

Figure 1 shows the sub-millimetre region relative to the total downwelling longwave spectrum for two polar locations. At
frequencies where the curve resembles the shape of a Planck function the atmosphere is opaque, whereas in atmospheric window regions, such as around 15,000 GHz, 27,500 GHz and 35,000 GHz, the radiance drops. Note that radiance is the radiant flux received per unit solid angle and is hence a directional quantity. The total energy received at any point will include contributions from all directions. The sub-millimetre's fractional contribution to the total energy incident at the ground from the nadir direction is small (below 0.5%), but not insignificant, accounting for a larger proportion of the total radiance in colder
and drier conditions (black line). To put this into context the downwards flux resulting from an isotropic radiance field (Flux = $\pi$Radiance) for both of these cases is about $0.6\,\mathrm{Wm}^{-2}$, which is approximately equal to the total energy imbalance of the Earth (Smith et al., 2015; Hansen et al., 2011). For maximum accuracy across the whole longwave spectrum the sub-millimetre would benefit from the same rigorous treatment applied to other regions to provide confidence that it is sufficiently well represented by radiative transfer algorithms in line-by-line and global climate models. Progress in this regard is the result of radiative closure
studies which are designed to 'minimise differences' between models and observations e.g. (Delamere et al., 2010; Turner et al., 2004; Clough et al., 1994). In brief, these studies involve making spectral measurements across the frequency range of interest whilst simultaneously characterising the surrounding atmospheric state, often with concurrent radiosonde launches, the data from which are used to drive radiative transfer calculations to compare with the observed radiances. Residual analysis is then used to improve the models by adjusting spectroscopic and continuum parameters to better fit the measurements. In clear-
sky simulations the main focus of these experiments at frequencies in opaque regions is to test water vapour absorption and





emission parameters, both line related and continua, the latter of which is known to dominate model uncertainties (Delamere et al., 2010).

The continuum contribution is what remains after deducting the radiance produced by allowed spectral transitions, which takes the form of a slowly-varying function with respect to frequency (Clough et al., 1989). The strongest effects are seen in

the atmospheric windows, where the continuum is often the main contributor to the total radiance. The functional form and underlying theory remain an area of scientific uncertainty (Shine et al., 2012) and as such it is parameterised semi-empirically in radiative transfer models in order to provide agreement with in-situ measurements rather than resolving the underlying processes explicitly. Hence, in order to improve and validate its representation at all frequencies radiative closure studies are required in every spectral region in which water vapour is radiatively active. To date, there have been water vapour continuum

closure studies in the infrared (Mlawer et al., 2012), far infrared (Fox et al., 2015; Masiello et al., 2012; Delamere et al., 2010; Tobin et al., 1999), microwave (Wentz and Meissner, 2016; Payne et al., 2011; Turner et al., 2009) and a single study in the sub-millimetre region between 350 - 1,100 GHz (Pardo et al., 2001b). To our knowledge, the equivalent has not been performed in the region between 1,100 - 10,500 GHz. Terahertz continua have been measured under controlled conditions in the laboratory (Slocum et al., 2013; Podobedov et al., 2008), however, the exact atmospheric conditions and composition of the air is difficult

to reproduce without in-situ measurements. Therefore continuum models at sub-millimetre wavelengths are determined by analytical continuation of values determined in other regions. Given the large uncertainty in these extrapolated values, which will be exacerbated with altered water vapour concentrations, such as the increased amounts in the upper troposphere predicted under climate change (Chung et al., 2014), radiative closure studies would be a highly beneficial test of the representation of this spectral band.

In addition to ice clouds and water vapour, the sub-millimetre atmospheric spectrum is rich in lines arising from primarily the pure rotational transitions of many different atmospheric gases, some of which do not have observed lines at other wavelengths. Though many of these signals are weak against the background atmosphere compared with those in the infrared for example, advances in technology may offer the potential for retrieving vertical gas concentrations with equivalent or better accuracy than previously achieved, even from the ground. Selecting the optimum line (or lines) to characterise the distribution of a

particular molecule is a vital step which can then be used to provide a framework for developing future instruments. Previous examples of this approach are: Kasai et al. (2006) who simulated the observational capabilities of various sub-millimetre bands for identifying ozone isotopes prior to the past SMILES mission, Manago et al. (2014) who scoped BrO and HOCl for the planned SMILES-2 mission and (Urban, 2003) who determined the optimum signals of HBr, BrO, HOCl and $HO_2$ for potential future satellite missions, as did Jiménez et al. (2007) and Buehler et al. (2007) with regards to cloud ice. For ground-

based remote sensing, Ryan and Walker (2015) simulate measurements of $HNO_3$, $O_3$, $N_2O$ and ClO for the proposed Arctic SPÉIR instrument which will operate in the frequency range below 300 GHz. Unlike these previous studies the present work does not apply retrieval algorithms to obtain the vertical distributions of each gas from the identified signals, which is instead left for future studies.





## 3 Methodology and Data

The downwelling spectrum at sub-millimetre wavelengths is dominated by the vertical distribution of pressure, temperature, water vapour, nitrogen, oxygen and ozone, which we term the 'background' atmosphere. Ozone is included because of its strong radiative properties in this frequency range. Constructing accurate background atmospheres is crucial when characterising particular locations, as these parameters can vary considerably between sites. The method adopted is described in section 3.1. Added to the background atmosphere are a further 24 gaseous species, including HOBr, HBr, $HO_2$ and $N_2O$, to produce what we term the 'complete' atmosphere, the sources of which are detailed in section 3.2. Climatologies of these 30 parameters are constructed for input to a line-by-line radiative transfer forward model, which is configured as described in section 3.3. Simulated downwelling brightness temperatures and transmissions are produced at a sampling resolution of 0.5 GHz initially to perform a survey of the whole sub-millimetre region. To isolate the radiative contribution of each species of interest an equivalent simulation is performed which omits it, so the residual between the two sets of radiances provides an estimate of the atmospheric 'signal' of the species separated from the signal from the rest of the model atmosphere. The full spectral surveys provide a guide to selecting target lines based on the signal strength of the source molecule, which are then analysed in more detail by performing a further simulation at a higher frequency resolution in the narrow region surrounding the line. The procedure followed to determine measurement characteristics for each line identified is described in section 3.3.1.

### 3.1 Background Climatologies

An overview of the sites chosen at which downwelling radiances are simulated is given in Table 1, along with the reasons for their selection. Broadly, these reasons are either the sites' high elevation and/or cold conditions giving the benefit of low water vapour concentrations at ground level and in the troposphere, and also the availability of existing science infrastructure at the site. Particular care is taken in constructing profiles as features such as irregular orography and temperature and humidity inversions, which are particularly prevalent in the four high latitude sites selected (Zhang et al., 2011), can have a large effect on sub-millimetre radiative transfer. Modelled quantities calculated on a fixed grid of locations do not accurately reproduce the level of detail captured by in-situ observational systems.

For all sites, radiosonde data are obtained from various previous campaigns, apart from Mauna Kea in Hawaii where only ground level meteorological data are available. For Halley and Rothera, separate surface synoptic measurements taken at the time of balloon launches are used for the lowest level of each radiosonde profile. For diurnal consistency across all sites data are sub-selected to retain only local noontime measurements or, where data availability is limited, as close as possible within a 3 hour time window. For the Greenland campaigns radiosondes were regularly launched between 11:00 and 12:00, for the Antarctic locations times tended to be between 10:00 and 12:00, for Mauna Kea 12:00 was achievable daily as the weather station monitors recorded continuously, and the majority of balloon launches from Atacama were in the afternoon from 13:00. The time period covered by each dataset is shown in Figure 2. All available data are used to construct profiles at each site to obtain the most typical representation of atmospheric conditions possible.



As many of the species of interest are present in significant abundance in the stratosphere and above, which is beyond the coverage of most in situ observations, ERA Interim reanalysis data (Dee et al., 2011) are spliced on top of each available radiosonde profile of water vapour and temperature. This is the latest global atmospheric reanalysis produced by the European Centre for Medium-Range Weather Forecasts (ECMWF) which is calculated at a T255 spectral resolution (~0.7°latitude x

~0.7°longitude) at 60 vertical pressure levels up to 0.1 hPa. The temporal resolution is six hourly in Coordinated Universal Time (UTC). The ERA Interim profile associated with each radiosonde launch is the one closest to noon, after adjusting UTC for each site's local solar time. For the selected sites this is at 18:00 UTC, apart from Mauna Kea which is 00:00 UTC.

The water vapour profiles obtained using only radiosonde measurements and only ERA Interim data are compared in Figure 3. For the high-altitude sites, the Atacama Desert, Mauna Kea and Summit, the positions of the nearest ERA Interim gridpoint

are at lower altitudes, yielding water vapour concentrations that are too high at ground level. Radiosonde data at pressures below 100 hPa seem to show spuriously high humidity values, the precise source of which is not clear. However, it is well known that radiosonde humidity measurements are less accurate at low temperatures (Wang et al., 2002). One reason for this inaccuracy is ice deposition on the sensor in an ice-supersaturated environment, which can lead to suspiciously elevated measurements well into the stratosphere (Miloshevich et al., 2006). Therefore, the final spliced profile combines observational data at lower

levels up to a threshold of 100 hPa and ERA Interim between 100 hPa and 0.1 hPa, with the exception of Mauna Kea profiles which are comprised entirely of reanalysis data apart from the lowest altitude level. Ozone profiles are derived solely from ERA Interim, which is sufficient because concentrations in the troposphere are very low compared to the stratospheric ozone layer. Molecular oxygen and nitrogen are set at fixed values of 209 000 ppmv and 781 000 ppmv, respectively, as they are well mixed and show little variability below 64 km (0.1 hPa).

### 3.1.1 Scenarios

Spliced profiles are generated for each site under six different atmospheric scenarios, which are described in Table 2. For the low humidity scenario (<10% PWV) precipitable water vapour is calculated as the sum of the mean water vapour density over all layers at altitudes provided by the radiosonde data. For Mauna Kea altitude information was not available so an estimate of PWV was made based on the method described by Smith (1966). As the PWV values are used solely for partitioning the

25 data high absolute accuracy is not necessary. Figure 4 shows water vapour profiles, constructed as described in the previous section, for all scenarios at each site. All constituent profiles have a local time close to noon, hence the equatorial locations, Atacama and Mauna Kea, have no Night scenario because there is little seasonal variation in daylight. Therefore the Day and the Annual mean scenarios are one and the same. These scenarios have been chosen to incorporate the range of potential science interests of the molecules studied, some of which have strong seasonal variations at the high latitude locations, and

30 strong diurnal cycles due to processes such as photolysis. For example, in the polar locations, humidity inversions are present in some of the scenarios shown in Figure 4, but not others, which will have a strong effect on the simulated radiances. The low humidity case provides an estimate of the highest trace gas signal strengths that would occur 10% of the year, which may be useful for measuring molecules with signals around the detection limit where a climatology rather than continual monitoring is sufficient.



## 3.2 Trace Gas Climatologies

All trace gases included in the complete atmosphere and their data sources are listed in Table 3. Certain brominated species such as HBr and HOBr have been sparsely measured so, for all four species of interest, climatologies are constructed from atmospheric chemistry climate model data. The model used is the stratospheric chemistry configuration of the UK Met Office Unified Model - UK Chemistry and Aerosol (UM-UKCA) (Morgenstern et al., 2009) which has an N48 horizontal resolution (2.5°latitude x 3.75°longitude) and 60 vertical levels extending up to 84 km. The chemistry scheme has recently been updated with tropospheric bromine chemistry based on work with the pTOMCAT chemistry transport model (Yang et al., 2014, 2010, 2005). Data output at two UTC times: 01:00 and 13:00, for the whole of 2012, are adjusted to local solar times for each site and, as with ERA Interim, the time closest to noon is retained. The zonal mean distributions of vertical concentration show considerable variation between each of the four species (Figure 5). HOBr and HBr are the least concentrated with units on the order of parts per trillion by volume (pptv), $HO_2$ is three times more abundant than this with units of parts per billion by volume (ppbv) and $N_2O$ is six times more concentrated at parts per million by volume (ppmv). The first three of these molecules have strong seasonal cycles; statospheric HOBr abundance peaks in the polar winter due to prolonged periods of darkness, tropospheric HBr peaks over sea ice in the polar spring, and $HO_2$ peaks in the polar summer during the prolonged sunlit period. The UM-UKCA data provides an estimate of the variability of the species when divided into the six scenarios described in the previous section.

Additional gas climatologies are obtained from the Monitoring Atmospheric Composition and Climate (MACC) reanalysis dataset (Inness et al., 2013). The MACC reanalysis data are produced with the Global and Regional Earth System Monitoring Using Satellite and In Situ Data (GEMS) Integrated Forecast System (IFS cycle 36r1) model at ECMWF and have the same spatial and temporal resolution as ERA Interim. MACC includes fields for $CO_2$, CO, $H_2CO$ and $SO_2$ and has been validated against multiple sources of ground-based measurements and satellite data (Inness et al., 2013). As with the UKCA profiles all scenarios are derived from a complete 2012 dataset. Profiles for remaining species with significant sub-millimetre lines are obtained from the Air Force Geophysics Lab (AFGL) Atmospheric Constituent Profiles (Anderson et al., 1986). These are single globally averaged profiles from 0 to 120 km that are appropriate for U.S. Standard conditions (NOAA, 1976), and are derived from a variety of sources including global satellite measurements and models. In general, AFGL adopts daytime estimates for diurnally varying species.

## 3.3 Simulated downwelling observations

The Atmospheric Radiative Transfer Simulator (ARTS) (version 2.2.41) available at http://www.sat.ltu.se/arts/ is the forward model used in this study (Buehler et al., 2005; Eriksson et al., 2011). ARTS is a line-by-line model that can simulate radiances from the infrared to the microwave, and has been validated against other models in the sub-millimetre spectral range (Melsheimer et al., 2005). It includes contributions from spectral lines and continua via a choice of user-specified parametrisations. For our work, we use the Planck formalism for calculating brightness temperatures and spectroscopic line parameters are taken from the HIgh resolution TRANsmission (HITRAN) molecular database 2012 (Rothman et al., 2013). Other spectro-





scopic databases available include the JPL molecular line catalog, which was designed specifically for millimetre/submillimetre astronomical applications (Pickett et al., 1998). However, the JPL catalog does not include any pressure broadening parameters, which are vital when considering line overlap effects in the atmospheric spectrum. Preliminary investigations show that root mean square intensity differences between the two databases at corresponding lines for HBr, $HO_2$ and $N_2O$ are within 4%

for the 10% highest line intensities in the sub-millimetre range. The error in the conversion between JPL and HITRAN line intensities is within 2%. The HOBr line intensities show far greater mean differences, of the order 71%, because unlike the JPL catalog, HITRAN does not resolve the hyperfine structure resulting from the intramolecular electromagnetic interactions of the nuclei with non-zero spins (Koga et al., 1989). Each hyperfine group in the JPL line list is composed of four very close transitions at frequencies within a 1-6 MHz range, with the sum of the component intensities agreeing with HITRAN to within

4%. As we use the HITRAN representation of these single, rather than split, HOBr lines this will potentially impact the radiative transfer calculations by overestimating signal strengths. It should also be noted that HITRAN does not include the $N_2O$ lines between 1,357 - 15,456 GHz, included in the JPL database.

The water vapour continuum parameterisation used is the most recent version of the Mlawer-Tobin Clough-Kneizys-Davies (MT-CKD) model (version 2.5.2), which separately includes both foreign and self broadening components (Mlawer et al.,

2012). Other possible choices investigated are an earlier version of the MT-CKD model (version 1.0) (Clough et al., 2005), which does not include more recent adjustments to the continuum coefficients based on observation fitting, and an even earlier formulation, CKD (version 2.4) (Tobin et al., 1999), based on (Clough et al., 1989), which does not include collision-induced modifications. The Ma and Tipping (2002) model is the only fully theoretical model included, which is formulated from first quantum mechanical principles without fitting parameters to experimental data. Additionally, we test the Millimeter wave

Propagation Model (MPM93) (Liebe et al., 1993) which includes both self and foreign continua and is commonly used for simulating the water vapour continuum at lower frequencies, and the Rosenkranz (1998) model which combines the self broadening component from MPM93 and the foreign component from the earlier Liebe (1989) formulation. Collision induced absorption (CIA) is the main contribution to the dry continua in the sub-millimetre, hence for $N_2$, $O_2$ and $CO_2$ the CIA parameterisation from the MT-CKD model (version 2.5.2) (Clough et al., 2005) is applied. All simulations are performed for a

single pencil beam of radiation at a zenith angle of $0°$ which is defined as looking straight up from the ground.

### 3.3.1 Estimation of receiver characteristics

To measure a species signal it must be sufficiently distinguishable from the underlying atmospheric spectrum. The limit of detectability is set by the background noise, which arises due to the inherent natural variation of the flux of photons arriving at the receiver (Benford et al., 1998). The statistical fluctuation $\Delta T$ (K) in the total system temperature $T_{sys}$ (K) is described by

the ideal radiometer equation (Kraus, 1966):

$$\Delta T = \frac{T_{sys}}{\sqrt{\Delta \nu \tau}} \tag{1}$$

where $\Delta \nu$ is the signal bandwidth (Hz), $\tau$ is the integration time (s) and we define $T_{sys}$ (K) as:

$$T_{sys} = T_{atm} + T_{sig} + T_{rcr} \tag{2}$$





where $T_{atm}$ is the atmospheric brightness temperature of the underlying atmosphere (K), $T_{sig}$ is the signal strength (K) and $T_{rcr}$ is the receiver noise temperature (K). The signal to noise ratio ($T_{sig}/\Delta T$) is set to 2. This results in the following relationship between the $T_{rcr}$ and $\tau$ for a particular signal:

$$T_{rcr} = T_{sig}\left(\frac{\sqrt{\Delta\nu\tau}}{2} - 1\right) - T_{atm} \qquad (3)$$

The value of $T_{rcr}$ defines a minimum possible integration time. If instead the maximum integration time is set, this specifies a maximum receiver temperature which can be used to direct instrument design. The present study produces estimates of both quantities. To assess the minimum integration times that could be achieved, estimates of system noise temperatures for the various frequency bands are based on the front-end characteristics of the ALMA telescope receivers (Remijan, 2015), listed in Table 4. It should be noted that ALMA is a heterodyne system which represents the current state-of-the-art in receiver technology, however, in principle smaller $T_{rcr}$ will be possible with future developments. For the inverse situation we specify a typical integration time of 30 minutes, chosen because this is a typical timescale of atmospheric processes and matches the minimum timestep of many atmospheric models. This is additionally the time taken for a typical radiosonde ascent, which is advantageous when designing missions that require concurrent characterisation of the atmospheric state. If the bandwidth chosen encompasses the full frequency extent of the signal the measurement will produce the total integrated radiance without resolving the finer features. This result could be appropriate if the total column abundance of a species is required. However, if a stratified vertical profile is the desired outcome the bandwidth needs to be split into a number of channels to characterise the height-dependent pressure-broadened line profile of the gas, which in turn will alter the dependant variables $T_{rcr}$ and $\tau$ in equation 3. For consistency across all cases the bandwidth in this study is set to the full width at half maximum (FWHM) of the peak signal arising from the gas of interest.

# 4 Atmospheric simulations

## 4.1 Background atmosphere

Simulated downwelling spectra over the whole 10 - 2000 GHz range for the background atmosphere are shown in Figure 6 for all six scenarios at Summit, Greenland. For the benefit of both the atmospheric and astronomical communities brightness temperature and transmission (from the top of the atmosphere to the surface) are shown in the upper and lower panels respectively. Frequency bands identified as atmospheric windows are shown in grey in the lower panel, with the condition that transmission falls to below 10% to define boundaries. The spectral features of both brightness temperature and transmission are essentially mirror images of each other so only the former is used in the remainder of this study. Below 300 GHz in the microwave region windows are present that are relatively invariant to scenario, however at higher frequencies transmission is strongly dependent on atmospheric conditions. The scenarios with the driest conditions at Summit; <10% PWV, Night and DJF, allow windows to open at higher frequencies. Above 1000 GHz the atmosphere is almost totally opaque due to water vapour absorption and the brightness temperature approximates the equivalent blackbody temperature immediately above ground level.



Prominent features include the strong oxygen absorption bands around 60 GHz and 120 GHz, water vapour bands at 183 GHz, 325 GHz and 380 GHz, and the fine structure within windows which is predominately due to ozone. The JJA and daytime scenarios produce similar results because northern hemisphere polar locations experience perpetual sunlight in the boreal summer, so essentially JJA is the extreme of the daylight scenario. The same is true for DJF of the nighttime scenario.

For the Antarctic locations this pairing is switched as there is perpetual sunlight in DJF (austral summer) and perpetual night in JJA (austral winter). This pairing is not necessarily reproduced in the corresponding behaviour of trace gases, however. Figure 7 compares equivalent spectra at all six locations for the annual mean and the driest scenarios to show the range of behaviour. The Atacama desert and Summit show a similar opening of partial windows at high frequencies, above 1000 GHz, by virtue of their high elevations, even though temperature conditions and seasonal variability are very different at the two locations.

Similarly, the spectra at Halley, Antarctica and Thule, Greenland are similar even though they are situated at opposite ends of the globe as they are both coastal polar locations with similar climatologies.

### 4.1.1   Water vapour continuum

The effect of continuum parametrisation on the downwelling radiation is shown in Figure 8 for both a higher and a lower humidity case at Summit. Contributions are largest in the atmospheric windows away from regions densely populated with

water vapour lines. As the ambient humidity decreases more windows are available but the contribution in each is reduced in proportion with the lower water vapour concentration. In its entirety, the continuum contributes up to 40 K to the brightness temperature for the two cases shown (green line), which is far greater than the signals of all trace gases considered. Its fractional contribution to the total downwelling signal is up to 60% (in radiance units) in this region compared with those at higher frequencies, as the overall energy is low and the continuum contribution is inversely proportional to frequency. The continuum

is almost totally dominated by the foreign component (orange line) in the sub-millimetre which is not necessarily the case in other regions, for example in the mid infrared the self continuum component contributes more than the foreign component.

It is clear from Figure 8 that the choice of continuum model implemented in ARTS has a significant effect on the absolute brightness temperature. For example, the MPM93 model is used extensively when calculating atmospheric radiative transfer at frequencies below 300 GHz, however radiance differences between this and the MT-CKD 2.5.2 model can reach up to 20% at

frequencies around 1000 GHz. Both CKD 2.4.1 and MT-CKD 1.0 are part of the same family as the control model MT-CKD 2.5.2 and hence show its evolution over time. The MPM93 and Rosenkratz models were developed separately to the MT-CKD family, where the latter is based on parts of the former to provide better agreement with observations. Modifications between successive models are the result of mainly tuning coefficients to fit observations in measured frequency bands, which are then extrapolated to poorly understood regions like the sub-millimetre. Interestingly, the Ma and Tipping parametrisation, which is

the only model to calculate the continuum from first principles without fitting to observations, produces results that are closest to the recently developed semi-empirical MT-CKD 2.5.2 model. The range of values obtained based on this handful of models gives an estimate of the uncertainty in this component of the simulation, which is known to dominate other errors such as the treatment of lines or other parts of the calculation. As continuum models are designed for certain frequency regimes it is





important to select carefully depending on their strengths and be aware that none of the models considered are well validated in the sub-millimetre regime, particularly above 1000 GHz (Buehler et al., 1996).

## 4.2 Signals of HBr, HOBr, HO$_2$ and N$_2$O

Signal strengths over a broad frequency range are calculated as described in Section 3 for all species, scenarios and locations,
a subset of which are shown in Figure 9 for two scenarios at Summit. There is high variability between species, in terms of number and intensity of lines available, and also between scenarios within a single species. Though Summit is a very dry transmissive location, HOBr produces a very small change in brightness temperature, below 0.5 mK in every scenario. HBr has sparse lines in the atmospheric window regions with a maximum brightness temperature change of 1.5 mK under very dry conditions. HO$_2$ shows signals that are at least an order of magnitude higher and the strongest N$_2$O lines have peak brightness
temperature above 1 K in all conditions. All species show a marked shift in signal distribution as conditions become dryer and more windows open up to reveal stronger lines at higher frequencies, hence the strongest line in the annual mean scenario is in nearly all cases different to the strongest line in the low PWV scenario. For example, in Figure 9b HBr has a single prominent signal at around 500 GHz, however, in low humidity conditions two lines emerge at 1500 GHz and 2000 GHz (9f) that are about 50 times brighter. It should be noted that almost all surveys show negative values at some frequencies indicating regions
where absorption dominates over emission.

Enhanced resolution simulations are performed in the vicinity of the strongest lines in each scenario and the corresponding measurement requirements are calculated. The results for each gas are shown for Summit in Figures (10, 11, 12 and 13) and results for all locations are listed in Tables 5, 6, 7 and 8. All panels show the full frequency extent of the strongest signal until it falls below 10% of the peak residual brightness temperature, which in many cases encompasses further peaks of lower
intensity. Red lines show the differences in brightness temperatures that result from using the Ma and Tipping parameterisation of the water vapour continuum instead of the MT-CKD 2.5.2. These appear to be minimal in all cases. As a guide, the position and intensity of transitions in the HITRAN database are shown in the panels below each signal.

For HBr the peak centred at 500.648 GHz is the brightest for all but the driest scenarios, where a stronger double peak emerges at 1999.8 GHz and 2000.22 GHz (Figure 10). The 500.648 GHz emission is composed of six component lines which
are resolved close to the centre of the peak where narrow-band emission by stratospheric HBr dominates. As previously noted, selecting the bandwidth is non-trivial as information about the gas at different heights is contained in different parts of the signal. For example, HBr volume mixing ratio reaches a maximum in the lower troposphere, decreases with height and then increases sharply in the stratosphere. The tropospheric contribution is manifested in the pressure broadened wings of each emission line, and the stratospheric contribution emerges as the narrow spikes close to the line centre frequency.
Depending on the bandwidth chosen, either within a single spike or encompassing the full width of the pressure broadened line, different vertical profile information can be retrieved from the measurement. In the DJF and nighttime scenarios the narrow HBr peak arises from the higher stratospheric signal under low water vapour conditions, whereas the results for JJA and daytime show greater contributions from the troposphere, with a smaller signal corresponding to a wider bandwidth. In the 10% lowest humidity case (10f), the double peak is the envolope of nine components at 1999.9 GHz and three at 2000.22



GHz. Interestingly, the more numerous and intense lines at 1999.9 GHz do not produce as bright a peak as the latter because of the greater attenuation of the signal by water vapour at the lower frequency, as shown by Urban (2003), who calculates the 1999.9 GHz signal to be the stronger of the two when viewed from a tangent altitude of 20 km. This highlights the importance of performing the full radiative transfer simulation rather than inferring behaviour from catalogued line intensities or previous

reports of signals observed from different locations or platforms.

HOBr is most abundant in the stratosphere (see Figure 5a) and hence produces narrow brightness temperature peaks with small, broader contributions from the tropospheric component (Figure 11). It is has a strong diurnal cycle because it is photolysed by solar radiation and thus its volume mixing ratio increases during nighttime, evidenced by the brighter signals in Figures 11c, e and f (the latter because most low humidity profiles will be produced at night). Without a mechanism to destroy

it, HOBr will build up in the stratosphere and troposphere, leading to the enhanced signals and wider pressure-broadened lines seen in these three scenarios. Each of the peaks comprises two to three components; however, as detailed in the Section 3.3 each of these is a composite of the hyperfine structure not resolved in the HITRAN linelist and thus the actual signal, spread over a wider frequency interval, could have a lower peak.

$HO_2$ in the stratosphere and mesosphere will produce narrow peaks, with FWHM line-widths in the range ~100 kHz - 1

15  MHz (Figure 12). The peak brightness temperatures that are calculated for the $HO_2$ peaks are one to two orders of magnitude lower than those observed at 254.44 GHz by the National Radio Astronomy Observatory telescope at Kitt Peak, Arizona (31 °N, 110°W) by Clancy et al. (1994), which could be due to the combination of two factors. Firstly, the spectral resolution is set at 1 MHz in the present simulations whereas a 250 kHz resolution of the Kitt Peak observations would yield a sharper line with higher peak brightness temperature. Secondly, our input atmospheric profiles are averaged climatologies and $HO_2$ has

strong diurnal and seasonal variability and can be enhanced by orders of magnitude during energetic particle precipitation and atmospheric ionisation events associated with solar and geomagnetic storms. It is therefore quite possible that observed signals could often be higher than those simulated here.

A summary of the results for all locations is tabulated in Tables 5, 6, 7 and 8, with associated minimum measurement integration times and maximum receiver noise temperatures. The latter is based on an integration time of 30 minutes which, for

most of the brominated scenarios and some of the $HO_2$ cases, is insufficient time to detect the signal of the species above the background noise with a signal-to-noise ratio of two (cases indicated with a dash). As the simulations use averaged profiles, however, this does not rule out the possibility of detecting these species when they are present at higher abundances, as will occur in the real atmosphere. Minimum integration times are of the order of six months or longer for HBr and HOBr, reducing to around nine hours for $HO_2$ and below one second for $N_2O$. Even though concentrations of HBr (and to a lesser extent HOBr)

are elevated above Antarctica with respect to other latitudes (Figure 5b) this is not reflected in the relative signal strengths for Halley and Rothera as other factors, such as the higher water vapour concentrations in these locations, eclipse this effect. Greater $HO_2$ signal strengths are seen in the Antarctic austral summer (DJF), relative to those in the Arctic boreal summer (JJA) by a factor of four. Uniquely, these signals are even stronger than the corresponding low humidity case.



## 5 Conclusions

An updated review of the clear-sky sub-millimetre atmosphere has been presented, incorporating an overview of recent technological developments in the fields of astronomy that could potentially address outstanding challenges in this region. We have developed a methodology for simulating clear-sky sub-millimetre (10-2000 GHz) atmospheric spectra by combining an existing, validated radiative transfer code with careful construction of atmospheric climatologies for selected locations. We have demonstrated how the choice of spectral lines for ground-based remote sensing depends strongly on atmospheric conditions at the individual locations. Significant differences have been found in the spectroscopic line data in two of the main reference databases, and between different water vapour continuum models, highlighting the need for improved theoretical and experimental spectroscopic data for remote sensing and climate studies in the sub-millimetre region. The optimal frequencies for measuring HBr, HOBr, $HO_2$ and $N_2O$ from the ground have been determined and preliminary receiver characteristics calculated and tabulated for all considered locations and scenarios.

The brightness temperature, centre frequency, and width of the sub-millimetre spectral peaks produced by each of the four selected atmospheric trace gases show dependence on site, season and ambient water vapour concentration, re-enforcing the need to quantify the full possible variability in measurements prior to instrument design. This highlights the merits of designing instruments with multiple channels throughout the sub-millimetre range as it is beneficial be able to observe many lines at different frequencies simultaneously. The sample molecules characterised span a range of detectabilities. The bromine compounds prove particularly challenging to observe but are not impossible to detect given long enough integration times and sensitive enough instruments, such as those primarily developed for astronomical applications.

As the results of this work are based on radiative transfer forward model output a rigorous approach was adopted to estimate its major uncertainties and capture the true behaviour of the atmosphere. Atmospheric climatologies representative of real atmospheric conditions at different sites were created by sourcing in-situ observational datasets, local to each of the selected sites, to append to high elevation reanalysis profiles. Uncertainties in the simulation parameters have been investigated by comparing different spectroscopic line databases and continuum parameterisations. Though differences in line intensities between the HITRAN database and JPL catalog are small, the wider issues relating to inclusion of lines identified for HOBr and $N_2O$ suggests that these catalogues are in need of improvement to facilitate studies such as this one, particularly for gases with low atmospheric concentrations. Further laboratory work carried out by the spectroscopic community would give greater confidence in simulating atmospheric radiative transfer in the sub-millimetre region and its application to remote sensing and climate studies. Additionally, as the details of lower energy regions are often neglected in energy budget studies there is a dearth of radiative closure studies in the sub-millimetre, hence the large contribution by the water vapour continuum in this region remains largely un-validated. Filling this gap necessitates a hyperspectral instrument that measures across a broad range of frequencies to be either constructed or re-purposed by collaborating with the astronomical sciences.

A strategy has been established in this study to determine the measurement potential of various atmospheric gases at hitherto largely unexploited wavelengths. This strategy can be applied to other species of interest in the terrestrial atmosphere, of which there are many with spectral signatures in the sub-millimetre, including ice clouds, which have not been investigated in the



current work but could be incorporated in future studies using the cloud parameterisations available in radiative transfer codes. The initial results presented here can be taken forward by applying inversion algorithms to retrieve vertical gas concentrations and provide a framework to guide future instrument design. This work shows that it is in principle possible to do important atmospheric science from the ground, at polar latitudes, with astronomical instruments.

5 *Acknowledgements.* This work, carried out as part of the SPECTRO-ICE project at the University of Cambridge and the British Antarctic Survey (BAS), was supported by UK Higher Education Innovation funding for ET and RC. DAN and AJ were supported in part by the UK's Natural Environment Research Council. We thank Michael Simmons for his contributions to the SPECTRO-ICE project, Xin Yang (BAS) for producing UKCA model data, and Patrick Eriksson, Stefan Buehler and Jana Mendrok for providing guidance for using the ARTS radiative transfer model. Peter Kirsch provided computer support for running ARTS at BAS.



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



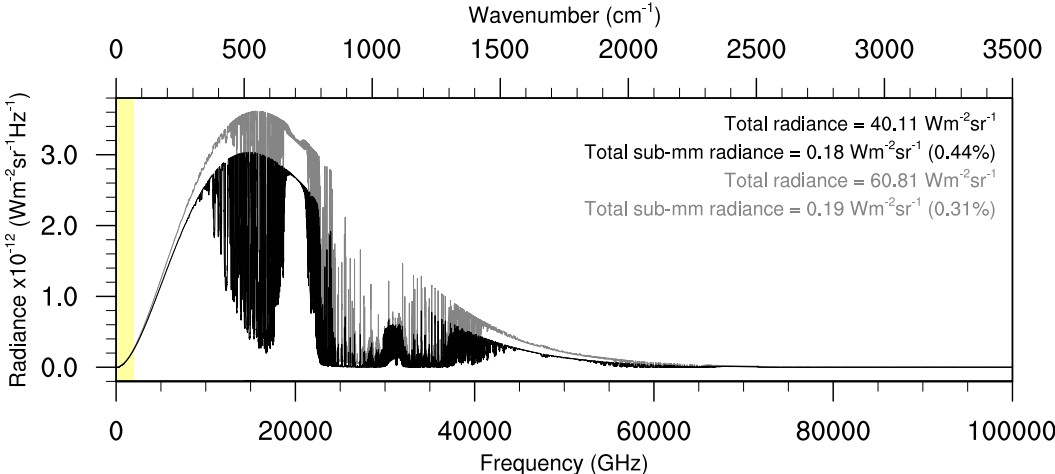

**Figure 1.** The simulated downwelling spectral radiance (zenith angle = $0°$) received at the Earth's surface over the entire longwave range (1 GHz sampling resolution). The sub-millimetre region (10 - 2000 GHz) is shaded in yellow. Calculations for two polar locations are shown:- a high, dry Arctic case (Summit, Greenland - 3216 m amsl, black) and an Antarctic coastal case just above sea level (Rothera, Antarctica - 16 m amsl, grey). For full details of the simulations see Section 3.



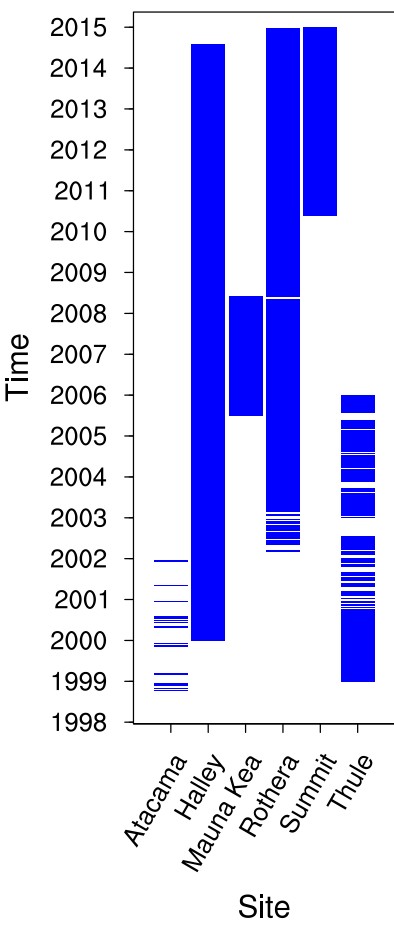

**Figure 2.** Time coverage of the background climatologies defined by the available radiosonde data from each site. Instrument details are given in Table 1.




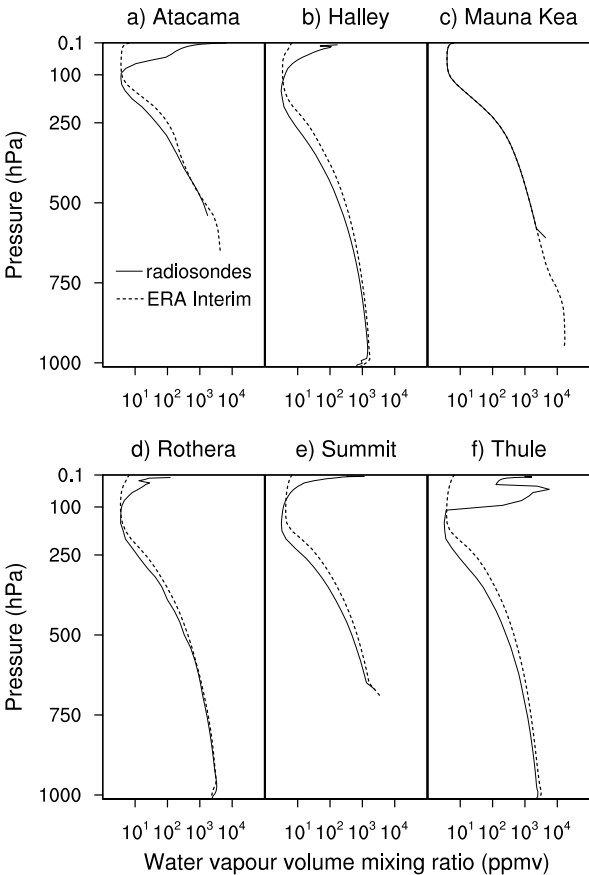

**Figure 3.** Climatologies of water vapour volume mixing ratio for radiosonde measurements (solid line) and ERA Interim reanalysis data (dashed line) at all six sites (a-f). Radiosonde profiles are the average of all data available at each site. ERA Interim data are sub-sampled to retain only the days when radiosondes were launched.





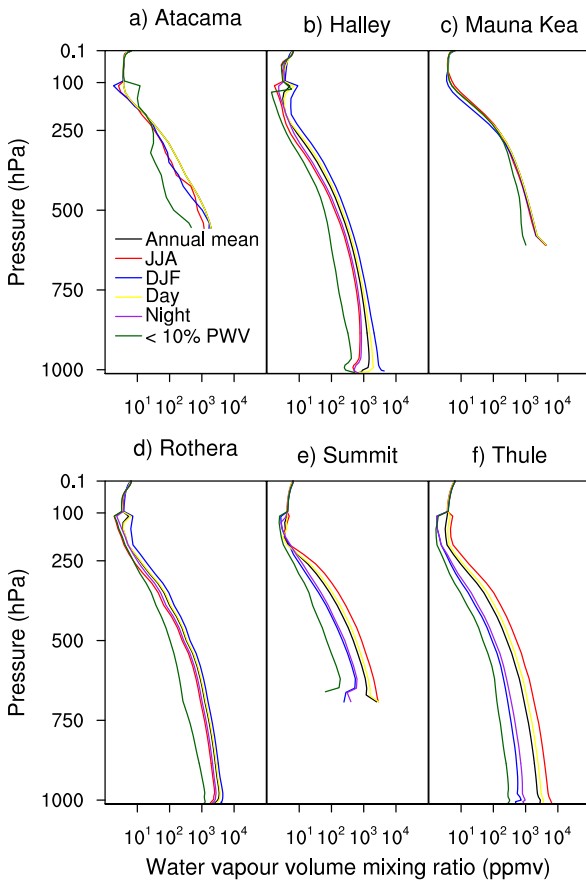

**Figure 4.** Climatologies of water vapour volume mixing ratio constructed with spliced radiosonde and ERA Interim data showing the six scenarios described in Table 2 for all sites (a-f).



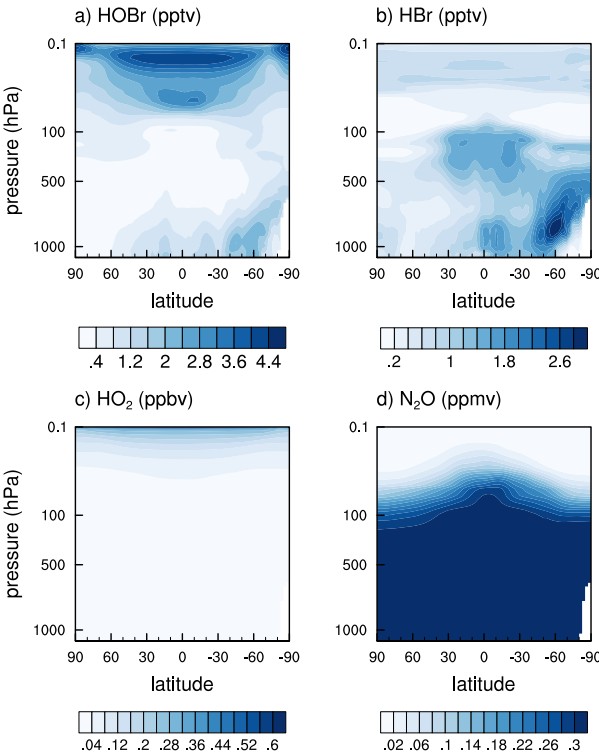

**Figure 5.** Zonal mean vertical concentrations for a) HOBr, b) HBr, c) HO$_2$ and d) N$_2$O produced by the stratospheric configuration of the UM-UKCA and interpolated to a fixed pressure grid. Data shown are the 2012 annual average. Note the different label bars and units.





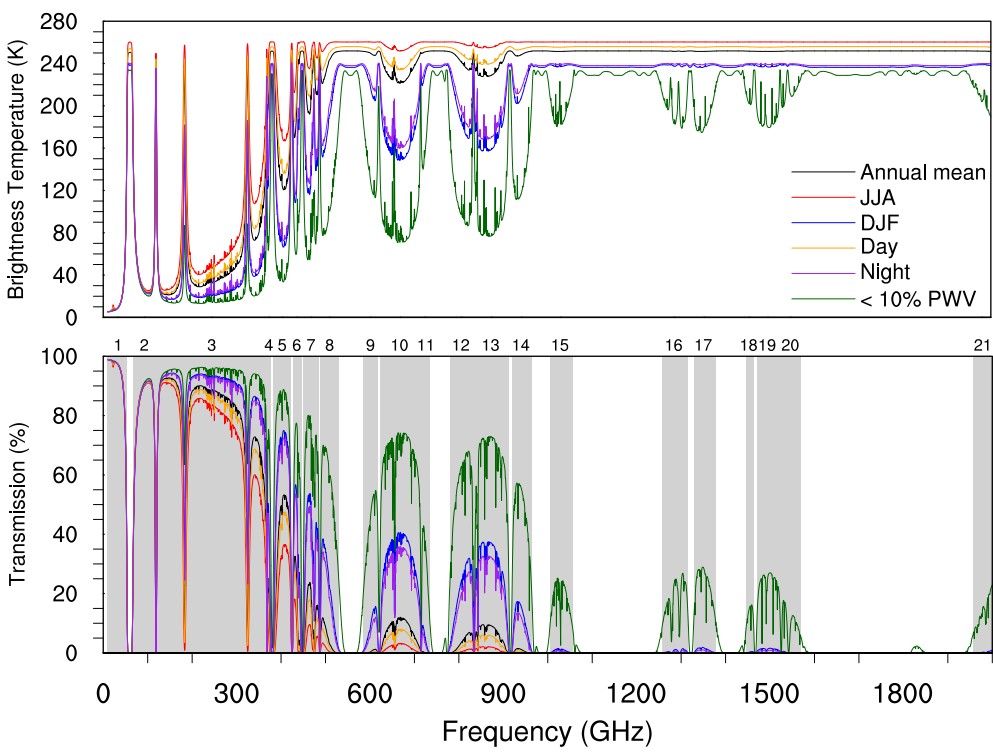

**Figure 6.** Downwelling brightness temperatures (top), and transmission (bottom) simulated by ARTS for the six scenarios, as viewed from the ground at Summit, Greenland. Window bands (full and partial) are shaded in grey on the lower panel and numbered above. The sampling resolution is 0.5 GHz.



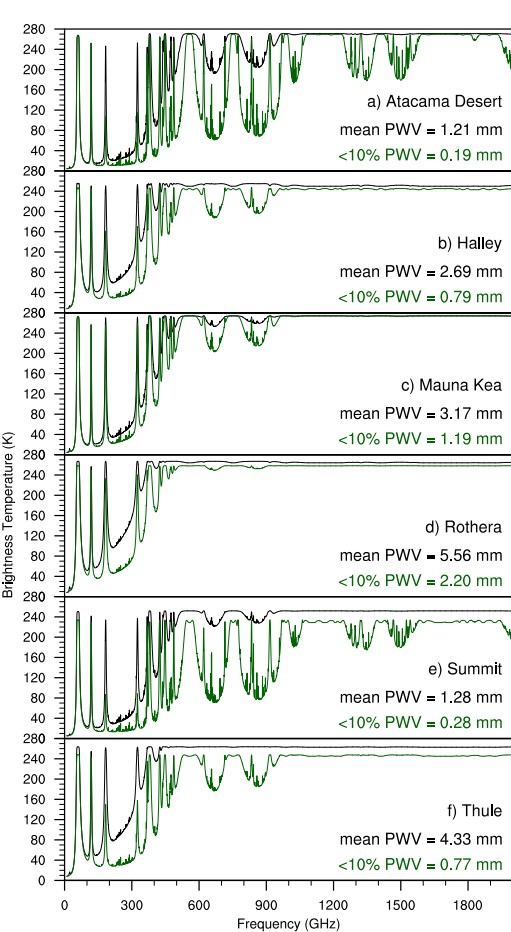

**Figure 7.** Simulated downwelling brightness temperatures as viewed from the ground for all six locations (a-f). The two scenarios shown are the annual mean (black curve) and the 10% driest profiles (green curve). The mean and the lower dectile PWV are shown for each site.





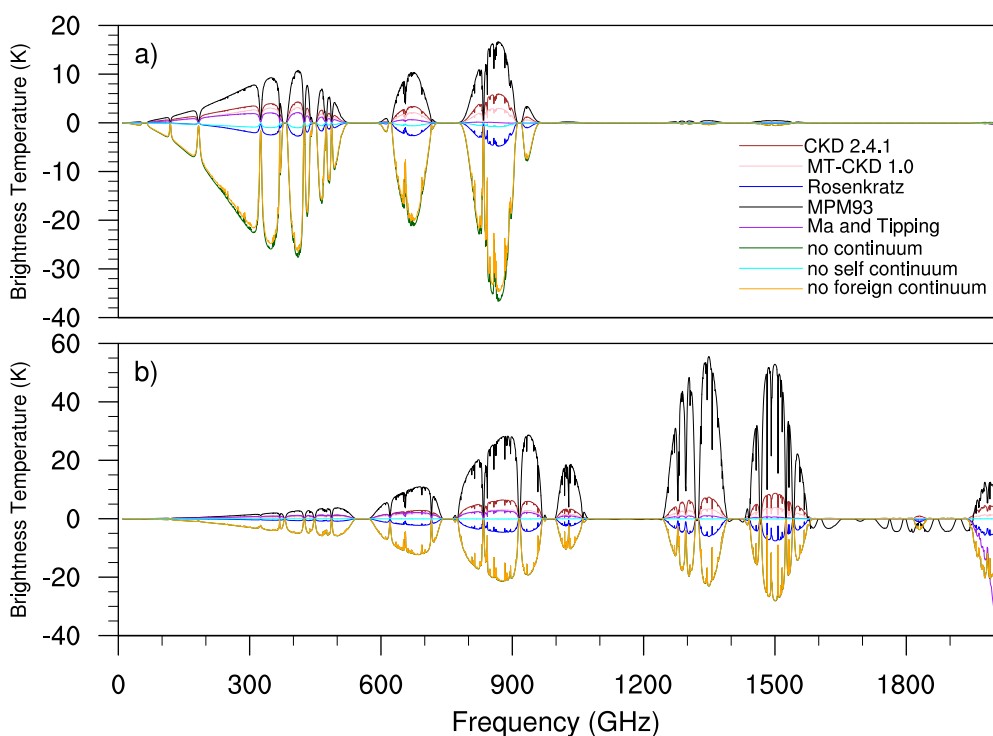

**Figure 8.** Spectral residuals between the downwelling brightness temperatures simulated with various continuum parameterisations available in ARTS and the configuration used throughout this study (MT-CKD 2.5.2) for a) annual mean, and b) <10% PWV scenarios at Summit, Greenland.





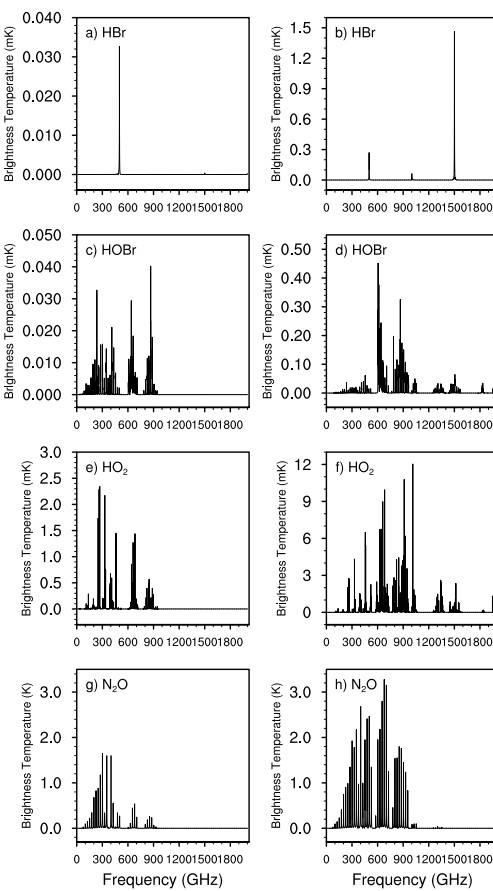

**Figure 9.** Total frequency range surveys of signal strength for the four species of interest at Summit, calculated as residual brightness temperatures between simulations of the complete atmosphere with and without the species (0.5 GHz sampling resolution). The plots on the left-hand side (a, c, e, and g) are for the annual mean scenario and those on the right-hand side (b, d, f, and h) are for the <10% PWV scenario. Units of the ordinate scale are milliKelvin (mK) except for the $N_2O$ plots which are in K. Note the different ordinate scales for each panel.





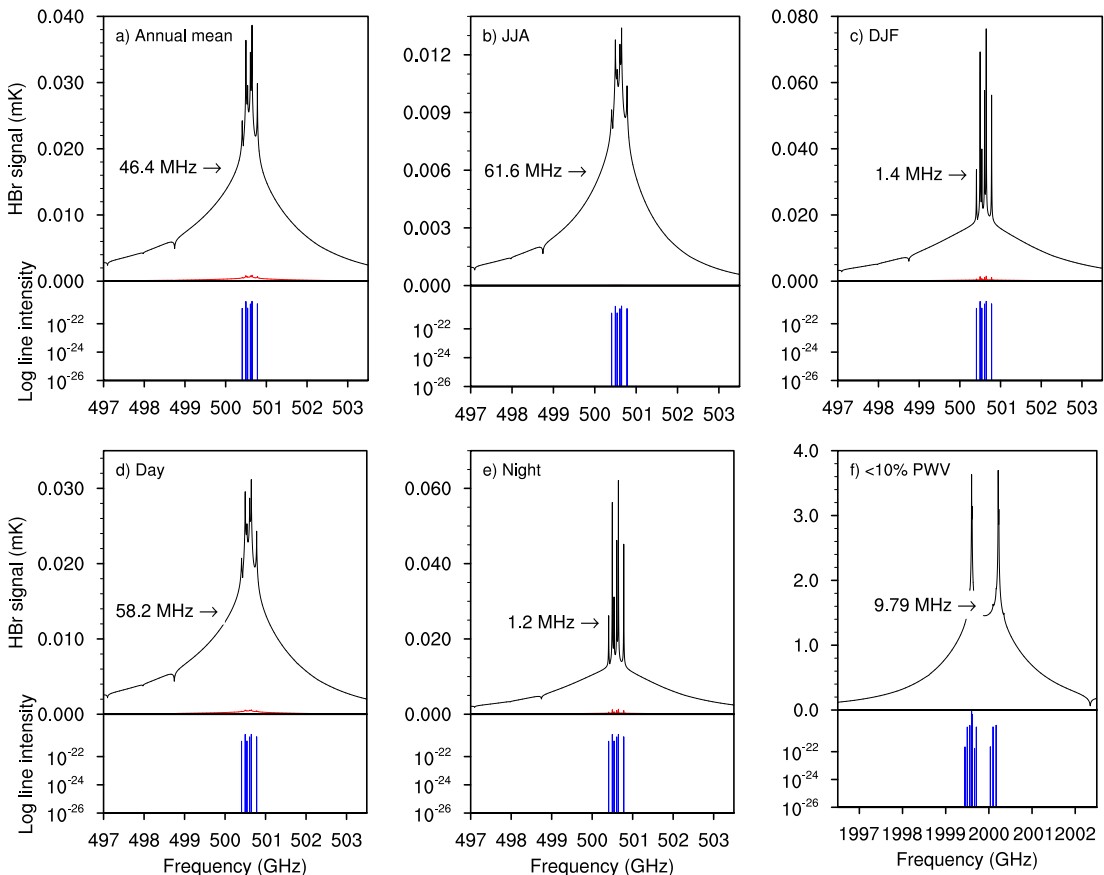

**Figure 10.** The enhanced HBr signal (1 MHz sampling resolution) around the strongest signal in each scenario (a-f) at Summit, calculated as a residual between simulations with and without HBr. Red lines show the differences that result from using the Ma and Tipping water vapour continuum model. Arrows point to the position of the FWHM of the peak signal in each plot, labelled with the corresponding bandwidth. Units of the ordinate scale are mK. The lower panels in each scenario show HBr intensities taken from the HITRAN database with $\log_{10}$ units of $cm^2$ (molecule $cm^2$).



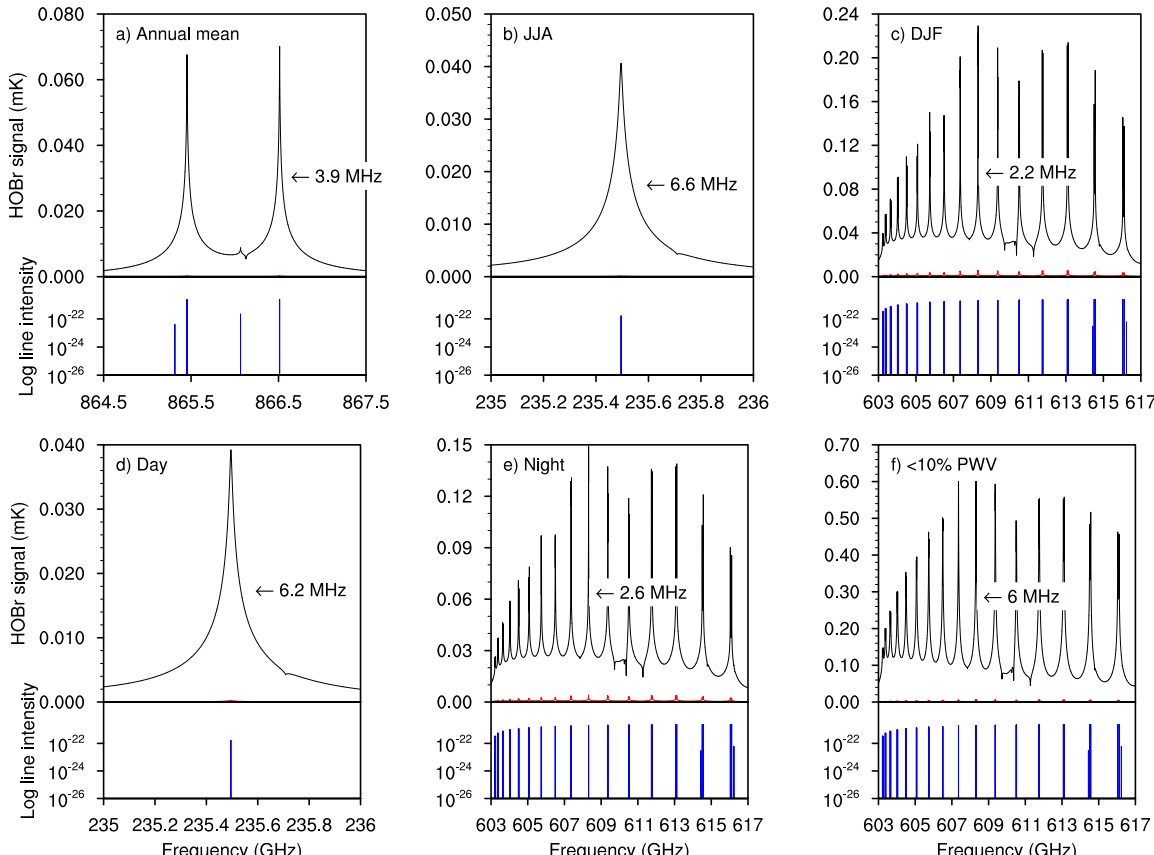

**Figure 11.** Same as Figure 10 but for HOBr.





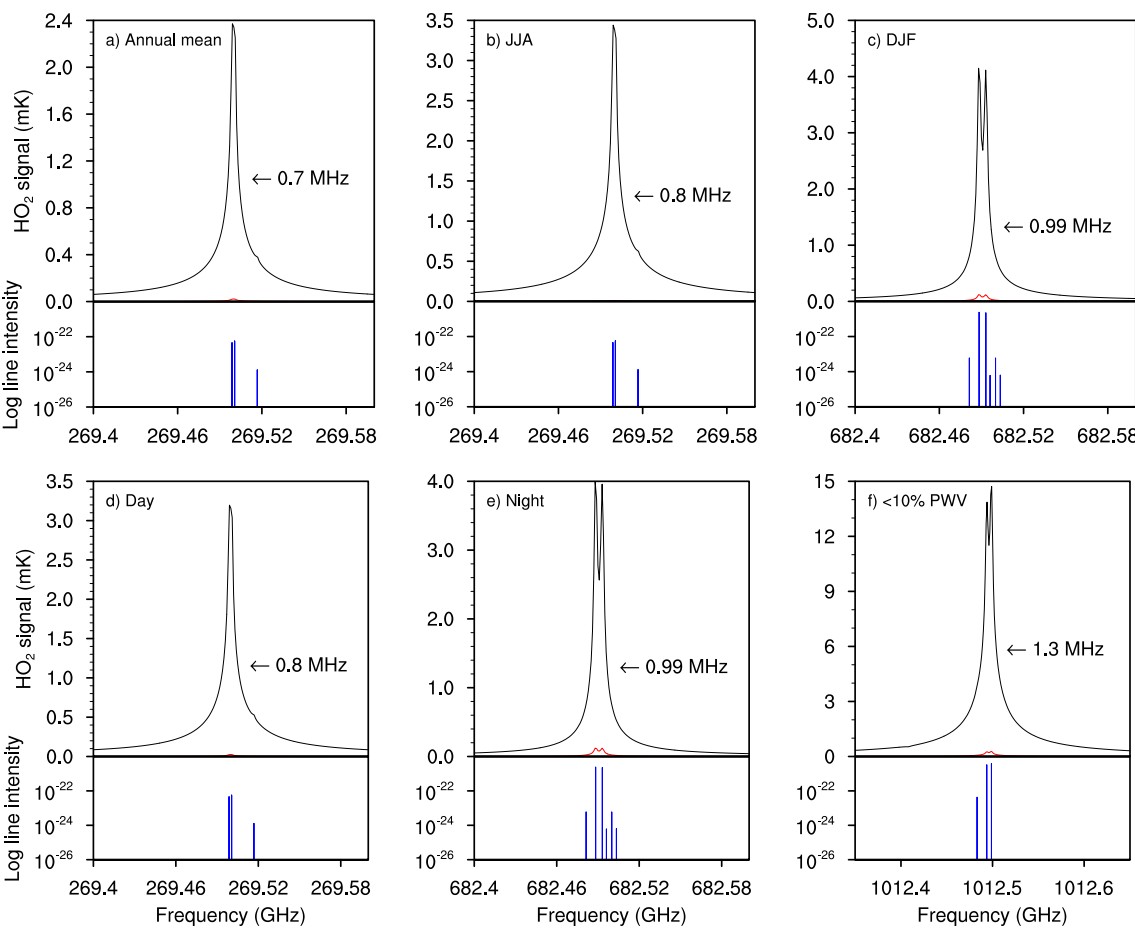

**Figure 12.** Same as Figure 10 but for HO$_2$.





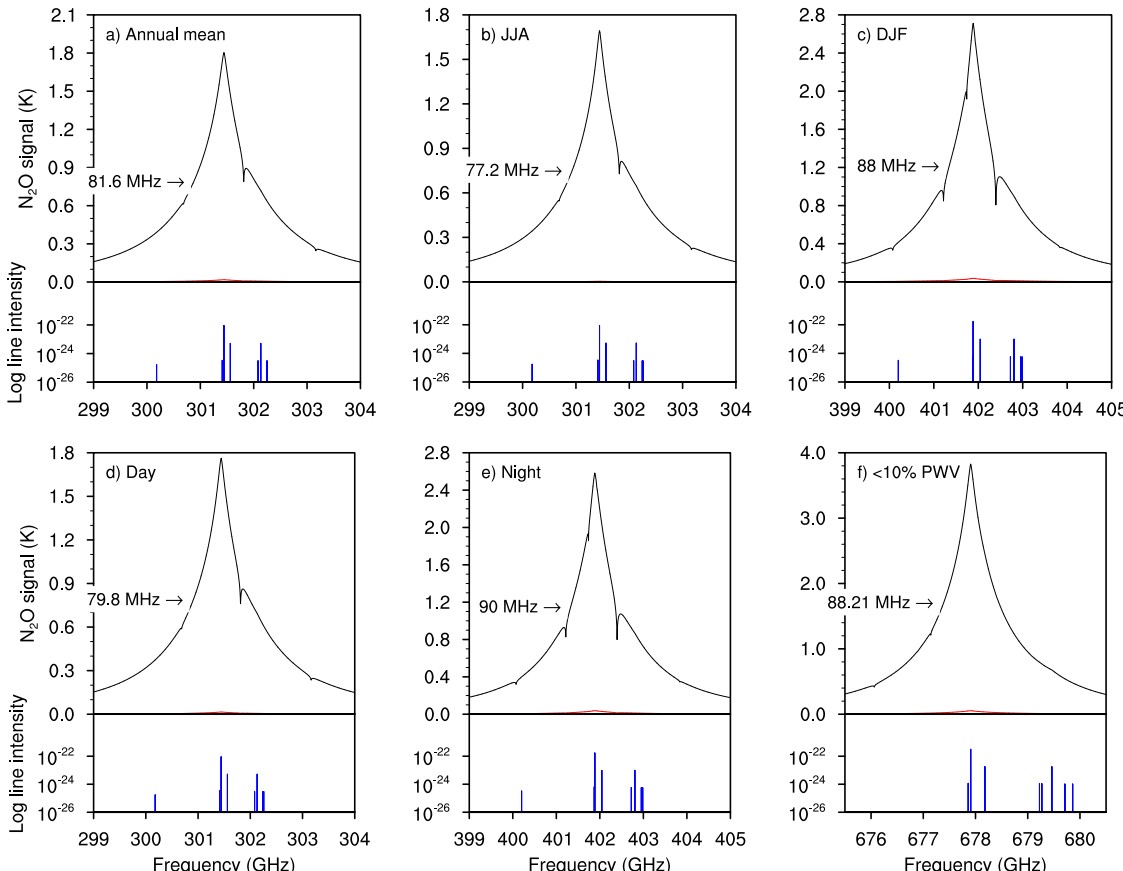

**Figure 13.** Same as Figure 10 but for N$_2$O. Units of the ordinate scale are K.



**Table 1.** Sites selected for sub-millimetre characterisation with observational sources.

| Site | Co-ordinates | Altitude (m) | Reason for selection | Observational Data |
|------|-------------|-------------|---------------------|-------------------|
| Atacama Desert, Chile | 23°1′9.42″S 67°45′11.44″W | 5055 | Site of the Atacama Large Millimeter/submillimeter Array (ALMA) radio-telescope which has a very high elevation and dry surroundings. | AIR 5A radiosondes from October 1998 - May 2001, replaced by Vaisala RS80 radiosondes from May - December 2001 as part of the Chajnantor radiosonde campaign (Giovanelli et al., 2001). |
| Halley, Antarctica | 75°36′16″S 26°12′32″W | 43 | British Antarctic Survey (BAS) research station located on an ice shelf. The site has cold and dry conditions. | Synoptic data from MAWSON weather station from 2000 - 2004, MILOS 520 automatic weather station from 2005 - 2006 and Campbell Scientific automatic weather station from 2007 - 2014. Vaisala RS80 radiosondes from 2000 - 2006 and Vaisala RS92 radiosondes from 2007 - 2014. |
| Mauna Kea, Hawaii | 19°49′14″N 155°28′05″W | 4205 | Site of the Mauna Kea Observatories which is home to 12 radio-telescopes at a high elevation. | Commercial automatic weather stations (AWSs) measurements made at 2 metre above the ground from Thirty Metre Telescope (TMT) site testing (Schöck et al., 2009). Produced by Monitor Sensors. |
| Rothera, Antarctica | 67°33′57″S 68°07′43″W | 16 | British Antarctic Survey (BAS) research station in a coastal location with cold maritime surroundings. | Synoptic data from MAWSON weather station from 2000 - 2004, MILOS 520 automatic weather station from 2005 - 2009 and Just Another Weather Station (JAWS) from 2010 - 2014. Vaisala RS80 radiosondes from 2000 - 2006 and Vaisala RS92 radiosondes from 2007 - 2014. |
| Summit, Greenland | 72°35′0″N 38°27′0″W | 3216 | Research station on the Greenland ice sheet with a high elevation and cold, dry conditions. | Vaisala RS92-K radiosondes from the Integrated Characterization of Energy, Clouds, Atmospheric State, and Precipitation at Summit (ICECAPS) program (Shupe et al., 2013). |
| Thule, Greenland | 76°31′52″N 68°42′11″W | 77 | U.S. Airbase and initial site of the planned GreenLand Telescope (GLT). Cold coastal conditions. | Vaisala RS80 VLF radiosonde launches compiled by the Met Office and archived with BADC (MetOffice, 2006). |

**Table 2.** The six scenarios constructed from all spliced radiosonde and ERA Interim reanalysis profiles. All constituent profiles are close to local noontime.

| Scenario | Description |
|----------|-------------|
| Annual mean | All profiles |
| JJA | June, July and August |
| DJF | December, January and February |
| Day | Elevation angle greater than 5° |
| Night | Elevation angle less than 5° |
| <10% PWV | Total column precipitable water vapour (PWV) less than or equal to the lower dectile PWV of all profiles at a particular site |





**Table 3.** Trace gases added to the background atmosphere (which includes $H_2O$, $O_3$, $O_2$ and $N_2$) and their climatological sources.

| Source | Constituents |
|---|---|
| UM-UKCA[a] | HOBr, HBr, $HO_2$, HCOOH, $CH_3OH$, $N_2O$ |
| ECMWF-MACC[b] | $CO_2$, $CH_4$, CO, $H_2CO$, $SO_2$ |
| AFGL[c] | NO, $NO_2$, $NH_3$, $HNO_3$, OH, HF, HCl, ClO, OCS, HOCl, HCN, $CH_3Cl$, $H_2O_2$ |

a. Unified Model - UK Chemistry and Aerosol (Met Office)

b. European Centre for Medium range Weather Forecasting - Monitoring Atmospheric Composition and Climate

c. Air Force Geophysics Lab

**Table 4.** Receiver noise temperatures for ALMA bands. Bands 3-10 are taken from Table 4.1 of Remijan (2015) and show the $T_{rcr}$ for any frequency in the range (the upper limit). Band 11 specifications are based on preparatory work by Masao Saito (Rigopoulou et al., 2013).

| Band | Frequency range (GHz) | $T_{rcr}$ (K) |
|---|---|---|
| 3 | 84.0-116.0 | <45 |
| 4 | 125.0-163.0 | <82 |
| 6 | 211.0-275.0 | <136 |
| 7 | 275.0-373.0 | <219 |
| 8 | 385.0-500.0 | <292 |
| 9 | 602.0-720.0 | <261 |
| 10 | 787.0-950.0 | <344 |
| 11a | 1000.0-1300.0 | ~500 |
| 11b | 1300.0-1600.0 | ~4000 |



**Table 5.** Characteristics of bands selected based on the strongest signal at each location and scenario for HBr. Minimum integration times, $\tau$, and maximum receiver noise temperatures, $T_{rcr}$, are calculated from equation 3. The $T_{rcr}$ used to calculate $\tau$ is given by the appropriate ALMA band in Table 4). $\tau$ is specified as 30 minutes when calculating $T_{rcr}$. The signal-to-noise ratio is 2. A dash indicates that the signal is below this threshold.

| Location | Scenario | Central frequency (GHz) | Atmospheric brightness temperature (K) | Signal strength (mK) | Bandwidth (MHz) | Minimum integration time (s) | Maximum receiver noise temperature (K) |
|---|---|---|---|---|---|---|---|
| **Summit** | all | 500.65 | 229.6 | 0.0387 | 46.402 | 1.549E+07 | - |
| | JJA | 500.65 | 255.1 | 0.0134 | 61.600 | 1.083E+08 | - |
| | DJF | 500.65 | 162.0 | 0.0762 | 1.404 | 1.011E+08 | - |
| | day | 500.65 | 240.4 | 0.0311 | 58.200 | 2.006E+08 | - |
| | night | 500.65 | 173.1 | 0.0621 | 1.202 | 1.878E+09 | - |
| | <10% PWV | 2000.22 | 186.2 | 3.6961 | 9.790 | 5.241E+06 | 59.166 |
| **Halley** | all | 500.65 | 254.8 | 0.0025 | 219.397 | 8.863E+08 | - |
| | JJA | 500.65 | 244.4 | 0.0106 | 20.801 | 4.887E+08 | - |
| | DJF | 500.65 | 266.5 | 0.0004 | 223.996 | 4.547E+10 | - |
| | day | 500.65 | 258.5 | 0.0011 | 239.001 | 4.514E+09 | - |
| | night | 500.65 | 248.0 | 0.0050 | 2.600 | 1.809E+10 | - |
| | <10% PWV | 500.65 | 200.2 | 0.2090 | 199.200 | 1.114E+05 | - |
| **Atacama** | all | 500.65 | 204.1 | 0.2505 | 112.802 | 1.391E+05 | - |
| | JJA | 500.65 | 166.3 | 0.4281 | 132.999 | 3.447E+04 | - |
| | DJF | 500.65 | 180.8 | 0.2653 | 73.599 | 1.726E+05 | - |
| | <10% PWV | 2000.2 | 181.9 | 10.7013 | 115.198 | 5.303E+03 | 2254.614 |
| **Mauna Kea** | all | 500.65 | 259.1 | 0.0602 | 123.599 | 2.711E+06 | - |
| | JJA | 500.65 | 256.4 | 0.0752 | 112.201 | 1.716E+06 | - |
| | DJF | 500.65 | 255.6 | 0.0752 | 137.799 | 1.537E+06 | - |
| | <10% PWV | 500.65 | 202.9 | 0.3290 | 121.201 | 7.470E+04 | - |
| **Rothera** | all | 254.70 | 115.5 | 2.151E-05 | 456.201 | 1.199E+12 | - |
| | JJA | 500.65 | 263.2 | 5.292E-05 | 95.200 | 4.625E+12 | - |
| | DJF | 254.70 | - | - | - | - | - |
| | day | 254.70 | - | - | - | - | - |
| | night | 500.65 | 264.7 | 1.860E-05 | 24.399 | 1.468E+14 | - |
| | <10% PWV | 500.65 | 252.9 | 0.0201 | 220.398 | 1.335E+07 | - |
| **Thule** | all | 500.65 | 263.7 | 7.061E-05 | 89.401 | 2.771E+12 | - |
| | JJA | 254.70 | - | - | - | - | - |
| | DJF | 500.65 | 236.0 | 0.0185 | 1.000 | 4.239E+09 | - |
| | day | 500.65 | 267.7 | 1.184E-05 | 210.001 | 4.255E+13 | - |
| | night | 500.65 | 247.3 | 0.0057 | 1.202 | 3.021E+10 | - |
| | <10% PWV | 500.65 | 190.6 | 0.1100 | 48.999 | 1.570E+06 | - |



**Table 6.** As Table 5 but for HOBr.

| Location | Scenario | Central frequency (GHz) | Atmospheric brightness temperature (K) | Signal strength (mK) | Bandwidth (MHz) | Minimum integration time (s) | Maximum receiver noise temperature (K) |
|---|---|---|---|---|---|---|---|
| **Summit** | all | 865.45 | 228.1 | 0.0676 | 3.998 | 7.161E+07 | - |
| | JJA | 235.50 | 45.0 | 0.0406 | 6.599 | 1.229E+07 | - |
| | DJF | 608.32 | 207.1 | 0.2292 | 2.203 | 7.573E+06 | - |
| | day | 235.50 | 38.6 | 0.0392 | 6.200 | 1.278E+07 | - |
| | night | 608.32 | 215.7 | 0.1491 | 2.600 | 1.569E+07 | - |
| | <10% PWV | 608.32 | 116.2 | 0.6547 | 6.000 | 2.213E+05 | - |
| **Halley** | all | 235.5 | 65.7 | 0.0450 | 5.400 | 1.490E+07 | - |
| | JJA | 235.50 | 46.2 | 0.0066 | 0.800 | 4.002E+07 | - |
| | DJF | 233.80 | 84.1 | 0.0374 | 8.600 | 1.612E+07 | - |
| | day | 233.80 | 74.8 | 0.0372 | 9.399 | 1.365E+07 | - |
| | night | 300.91 | 77.6 | 0.100 | 0.800 | 4.391E+07 | - |
| | <10% PWV | 865.45 | 196.4 | 0.204 | 6.397 | 4.371E+06 | - |
| **Atacama** | all | 866.51 | 204.9 | 0.1922 | 5.695 | 5.728E+06 | - |
| | JJA | 866.51 | 166.9 | 0.2917 | 5.908 | 2.078E+06 | - |
| | DJF | 866.51 | 182.6 | 0.2725 | 5.298 | 2.820E+06 | - |
| | <10% PWV | 608.31 | 101.7 | 0.8563 | 6.396 | 1.122E+05 | - |
| **Mauna Kea** | all | 235.50 | 39.4 | 0.0466 | 5.701 | 9.954E+06 | - |
| | JJA | 235.50 | 37.1 | 0.0488 | 5.701 | 8.826E+06 | - |
| | DJF | 235.50 | 37.8 | 0.0440 | 5.701 | 1.093E+07 | - |
| | <10% PWV | 866.54 | 203.1 | 0.2022 | 5.908 | 4.956E+06 | - |
| **Rothera** | all | 235.5 | 105.5 | 0.0360 | 5.400 | 3.325E+07 | - |
| | JJA | 235.50 | 86.7 | 0.0558 | 0.999 | 6.385E+07 | - |
| | DJF | 233.80 | 122.6 | 0.0292 | 8.400 | 3.731E+07 | - |
| | day | 233.80 | 109.9 | 0.0303 | 8.600 | 3.057E+07 | - |
| | night | 235.50 | 92.9 | 0.0483 | 1.399 | 6.435E+07 | - |
| | <10% PWV | 235.50 | 51.6 | 0.0485 | 5.800 | 1.031E+07 | - |
| **Thule** | all | 235.50 | 86.7 | 0.0444 | 1.099 | 9.154E+07 | - |
| | JJA | 235.50 | 136.8 | 0.0211 | 6.000 | 1.111E+07 | - |
| | DJF | 235.50 | 43.8 | 0.1088 | 0.500 | 2.185E+07 | - |
| | day | 235.50 | 101.6 | 0.0242 | 5.499 | 6.986E+07 | - |
| | night | 235.50 | 50.0 | 0.0799 | 0.500 | 4.325E+07 | - |
| | <10% PWV | 865.45 | 186.0 | 0.2713 | 1.099 | 1.389E+07 | - |



**Table 7.** As Table 5 but for $HO_2$.

| Location | Scenario | Central frequency (GHz) | Atmospheric brightness temperature (K) | Signal strength (mK) | Bandwidth (MHz) | Minimum integration time (s) | Maximum receiver noise temperature (K) |
|---|---|---|---|---|---|---|---|
| **Summit** | all | 269.45 | 37.7 | 2.371 | 0.699 | 3.071E+04 | 4.350 |
| | JJA | 269.45 | 54.5 | 3.441 | 0.799 | 1.535E+04 | 10.728 |
| | DJF | 682.49 | 152.9 | 4.149 | 0.995 | 4.002E+04 | - |
| | day | 269.50 | 42.8 | 3.197 | 0.799 | 1.566E+04 | 17.820 |
| | night | 682.49 | 164.3 | 3.991 | 0.995 | 4.566E+04 | - |
| | <10% PWV | 1012.50 | 193.1 | 14.716 | 1.300 | 6.827E+03 | 162.735 |
| **Halley** | all | 260.57 | 72.1 | 8.969 | 0.400 | 5.386E+03 | 48.175 |
| | JJA | 269.50 | 55.9 | 0.463 | 0.598 | 1.149E+06 | - |
| | DJF | 260.57 | 99.8 | 12.852 | 0.400 | 3.368E+03 | 72.583 |
| | day | 269.50 | 87.1 | 2.697 | 0.800 | 3.421E+04 | - |
| | night | 269.50 | 59.7 | 0.476 | 0.598 | 1.130E+06 | - |
| | <10% PWV | 682.49 | 191.8 | 10.671 | 1.093 | 6.589E+03 | 44.866 |
| **Atacama** | all | 682.49 | 197.4 | 19.699 | 1.306 | 1.659E+03 | 280.140 |
| | JJA | 682.49 | 159.1 | 26.505 | 1.196 | 8.404E+02 | 455.735 |
| | DJF | 682.49 | 173.9 | 29.531 | 1.306 | 6.643E+02 | 541.990 |
| | <10% PWV | 1012.50 | 189.5 | 37.474 | 1.300 | 1.042E+03 | 716.869 |
| **Mauna Kea** | all | 260.57 | 43.6 | 18.843 | 0.400 | 9.088E+02 | 209.159 |
| | JJA | 260.57 | 40.8 | 20.538 | 0.400 | 7.413E+02 | 234.697 |
| | DJF | 260.57 | 41.7 | 17.291 | 0.501 | 8.442E+02 | 217.788 |
| | <10% PWV | 682.49 | 195.7 | 22.179 | 1.196 | 1.418E+03 | 318.859 |
| **Rothera** | all | 260.57 | 118.9 | 6.542 | 0.400 | 1.518E+04 | - |
| | JJA | 269.50 | 102.6 | 0.358 | 0.598 | 2.966E+06 | - |
| | DJF | 260.57 | 138.3 | 9.581 | 0.400 | 8.202E+03 | - |
| | day | 260.57 | 123.7 | 9.150 | 0.500 | 6.437E+03 | 13.627 |
| | night | 269.50 | 110.1 | 0.370 | 0.598 | 2.967E+06 | - |
| | <10% PWV | 260.57 | 54.9 | 9.372 | 0.500 | 3.315E+03 | 85.768 |
| **Thule** | all | 269.50 | 102.1 | 1.329 | 0.699 | 1.838E+05 | - |
| | JJA | 262.00 | 155.2 | 4.267 | 0.598 | 3.115E+04 | - |
| | DJF | 269.50 | 46.0 | 0.180 | 0.500 | 8.168E+06 | - |
| | day | 269.50 | 120.5 | 1.979 | 0.800 | 8.401E+04 | - |
| | night | 269.50 | 54.4 | 0.421 | 0.598 | 1.367E+06 | - |
| | <10% PWV | 682.49 | 180.9 | 9.872 | 1.093 | 7.337E+03 | 37.975 |





**Table 8.** As Table 5 but for $N_2O$.

| Location | Scenario | Central frequency (GHz) | Atmospheric brightness temperature (K) | Signal strength (K) | Bandwidth (MHz) | Minimum integration time (s) | Maximum receiver noise temperature (K) |
|---|---|---|---|---|---|---|---|
| **Summit** | all | 301.44 | 52.9 | 1.804 | 81.598 | 1.129E-03 | 292.745 |
| | JJA | 301.44 | 77.2 | 1.693 | 77.203 | 1.583E-03 | 240.327 |
| | DJF | 401.88 | 80.6 | 2.712 | 88.004 | 8.703E-04 | 459.122 |
| | day | 301.44 | 59.9 | 1.762 | 79.800 | 1.271E-03 | 272.040 |
| | night | 401.88 | 86.2 | 2.583 | 89.999 | 9.959E-04 | 433.610 |
| | <10% PWV | 677.91 | 75.2 | 3.828 | 88.208 | 3.578E-04 | 687.453 |
| **Halley** | all | 301.44 | 101.3 | 1.509 | 126.001 | 1.443E-03 | 258.065 |
| | JJA | 301.44 | 72.7 | 1.680 | 141.400 | 8.626E-04 | 351.096 |
| | DJF | 301.44 | 129.6 | 1.396 | 85.400 | 2.945E-03 | 144.027 |
| | day | 301.44 | 114.4 | 1.410 | 121.201 | 1.861E-03 | 214.881 |
| | night | 301.44 | 79.6 | 1.667 | 134.601 | 9.643E-04 | 330.642 |
| | <10% PWV | 401.88 | 107.8 | 2.505 | 110.199 | 9.362E-04 | 450.041 |
| **Atacama** | all | 401.88 | 102.1 | 3.439 | 45.599 | 1.172E-03 | 390.594 |
| | JJA | 401.88 | 80.0 | 7.781 | 48.001 | 1.985E-04 | 1063.559 |
| | DJF | 401.88 | 87.6 | 3.852 | 44.199 | 8.965E-04 | 455.7191 |
| | <10% PWV | 702.98 | 84.9 | 5.338 | 44.000 | 3.935E-04 | 666.281 |
| **Mauna Kea** | all | 301.44 | 65.1 | 2.325 | 50.201 | 1.905E-03 | 284.288 |
| | JJA | 301.44 | 61.1 | 2.403 | 49.200 | 1.780E-04 | 296.392 |
| | DJF | 301.44 | 62.3 | 2.291 | 52.997 | 1.827E-04 | 291.602 |
| | <10% PWV | 401.89 | 101.0 | 3.437 | 50.803 | 8.901E-04 | 418.642 |
| **Rothera** | all | 301.44 | 160.0 | 1.013 | 95.599 | 5.888E-03 | 50.103 |
| | JJA | 301.44 | 133.5 | 1.183 | 134.399 | 2.660E-03 | 157.448 |
| | DJF | 301.44 | 182.0 | 0.890 | 85.800 | 9.517E-03 | - |
| | day | 301.44 | 165.6 | 0.976 | 93.997 | 6.639E-03 | 35.184 |
| | night | 301.44 | 142.4 | 1.144 | 125.998 | 3.187E-03 | 130.046 |
| | <10% PWV | 301.44 | 77.5 | 1.794 | 121.402 | 9.114E-04 | 341.717 |
| **Thule** | all | 301.44 | 133.2 | 1.247 | 94.397 | 3.404E-03 | 123.826 |
| | JJA | 301.44 | 198.9 | 0.724 | 89.600 | 1.490E-02 | - |
| | DJF | 301.44 | 62.2 | 1.949 | 93.601 | 9.017E-04 | 337.858 |
| | day | 301.44 | 154.6 | 1.074 | 92.600 | 5.261E-03 | 64.568 |
| | night | 301.44 | 73.3 | 1.783 | 96.799 | 1.124E-03 | 298.869 |
| | <10% PWV | 401.88 | 102.2 | 2.706 | 101.398 | 8.486E-04 | 475.841 |