# Peer review of "Simulation of sub-millimetre atmospheric spectra for characterizing potential ground-based remote sensing observations"

_Atmospheric Measurement Techniques, 2016_

## Referee Comment (RC1) · Anonymous Referee #1 · 4 Jul 2016

This manuscript investigates the possibility to perform ground-based measurements of HOBr, HBr, HO2 and N2O at frequencies up to 2 THz. Considering the importance (and the expected increase) of bromine and N2O for atmospheric chemistry the aim of the manuscript is justified and it fits very well with the scope of AMT. Exploring the possibilities of ground-based measurements is especially important in the light of the expected lack of limb sounding measurements.

Radio astronomy makes use of recent technology development to obtain low receiver noise temperatures and the same technology can be applied for ground-based atmospheric sounding. The manuscript explores this option, and assess the best frequency window to use for each species, for different atmospheric conditions. My main con-

cern with the analysis is that only thermal noise is considered, all other error sources are ignored. This is particularly problematic as some of the measurements require a spectral accuracy of about 10 uK. Is it really possible to maintain a spectral purity of that level over weeks/months? Very small external disturbances could easily ruin the measurements. Could not even interference from astronomical sources be a problem (notice that stronger nearby transitions can be red- or blue-shifted, and end up on top of the target frequencies)? How should various disturbances be handled when averaging spectra?

It should be possible to make a rough characterisation of some additional error sources. Maybe most important is to check the interference of species giving stronger spectral features. I don't see anything in the analysis that catches if the target transitions are on top of e.g. ozone isotopologue transitions. If this is the case, variation in both overall ozone concentration and isotopologue fractioning could interfere strongly with the measurement, or even lead to false "detection".

Some error sources, such as reflections inside the receiver system, are hard to characterise in a general manner, but they should at least be commented. Could any such error source even be a "showstopper"?

The manuscript text is very well written. In fact I have no detailed comments worth mention.

---

## Referee Comment (RC2) · Anonymous Referee #2 · 18 Jul 2016

**General comments**

This paper presents a framework for determining the feasibility of ground-based sub-millimetre measurements of atmospheric trace gases at a particular site, and deciding which spectral lines are good choices for observation. The method uses estimates of instrumental parameters representative of current technology that is mainly used for astronomical purposes, and examples are given for six observation locations. The writing of high quality and the figures are clear and well created. I believe this work is suitable for AMT and will be of interest to many. I would recommend this work for publication if the following points can been addressed. The two biggest concerns are listed as the first comments.

Specific Comments

Page 10, line 24-25 A zenith angle of zero is not typical of a ground-based measurement. Usually measurements are performed at angles above 60 degrees or so. Viewing angle can have a significant effect on the signal-to-noise of a measurement, and depends on the relative strengths of the background atmosphere (i.e., absorption in the troposphere) and the signal. How will the presented results, for individual lines and continua, change when using a viewing angle that is more representative of a ground-based measurement?

Section 4.2 When calculating signal strengths in the enhanced resolution, and using this to estimate measurement times, etc., it matters whether the spectral lines of the gas of interest lie "on top" of (or overlap with) other spectral lines. This is particularly true if a small signal, like examples used here, overlaps with a strong signal, like that from ozone. If "strongest signal", as calculated here for a gas, is meant to indicate a good choice for an observation window if one wants to retrieve that gas, then it is quite important to screen for so-called interfering species. They can cause both random and systematic effects that can lead to poor choices of measurement windows. Has this been considered when choosing the best window for observation of a gas?

Page 4, line 10-22 This section of the introduction is focused on using the sub-millimetre part of the spectrum for remote sensing of clouds. That subject is not addressed in the paper at all and so its introduction seems of little relevance or to serve no purpose. If this is the case, it should be removed.

Page 5, line 4-18 This section of the introduction focuses on the planetary energy balance. Similar to the last point, that is not addressed in the paper and does not directly relate to the work presented. Without justification, this should also be removed.

Page 7, line 9 What happens in the event that the edge of a 0.5 GHz frequency bin lands within a spectral line signal, splitting it?

Section 3 Could you also specify the calculation grid that was used for the forward modelling?

Page 8, Line 32 "…10% of the year…" Where does this number come from?

Page 12, line 6-7 What is meant exactly by "the range of behaviour"? Shown are the driest and annual mean scenarios, which isn't representative of a full range. Please be specific here.

Page 13, line 14-15 Could you briefly elaborate on the origin of this effect?

Page 14, lin26-28 "As the simulations use averaged profiles, however, this does not rule out the possibility of detecting these species when they are present at higher abundances, as will occur in the real atmosphere." While "not rule out" may technically be true, it is possibly misleading. What kind of higher abundances do you mean? Would detection require a ten-fold, or hundred-fold, etc., increase in gas concentration? And is it reasonable to assume that this required concentration will occur in the atmosphere?

Page 14, line 34 By "Uniquely", do you refer to within this work, or in general?

Page 15, line 9-12 "The optimal frequencies for measuring HBr, HOBr, HO2 and N2O from the ground have been determined and preliminary receiver characteristics calculated and tabulated for all considered locations and scenarios." Related to the two main (first) comments: - Is this sentence valid if one were to make a measurement using a viewing angle that is more representative of a ground-based instrument? - If no account was made for the interference from spectral lines from other gases, then this sentence may not be true. Please consider and comment.

Figure 10 "The enhanced HBr signal". Do you mean an enhanced resolution simulation, or an enhanced HBr signal from higher gas concentrations? Please clarify.

Technical comments

Figure 6, lower panel The tick marks at label locations are missing (I don't know if this is intentional).

Figure 6 Please clarify what is meant by "bandwidth". It is ambiguous without clarification or reference to the section of main text.

Figure 6, 7, 8, 9, and 10 Particularly (or at least) for Figure 6 and 7, which relate to the discussion of windows "opening", the zenith angle for the simulation should be mentioned.

Table 2 For the description of Day and Night scenarios, could you please specify that it is the sun elevation angle? It can be slightly confusing as elevation angle is also used to refer to instrument pointing.

Page 3, line 6 I believe "catalyse" should be in plural form here as it refers to "family"?

---

## Author Comment (AC1) · 23 Sep 2016

This manuscript investigates the possibility to perform ground-based measurements of HOBr, HBr, HO2 and N2O at frequencies up to 2 THz. Considering the importance (and the expected increase) of bromine and N2O for atmospheric chemistry the aim of the manuscript is justified and it fits very well with the scope of AMT. Exploring the possibilities of ground-based measurements is especially important in the light of the expected lack of limb sounding measurements.

Radio astronomy makes use of recent technology development to obtain low receiver noise temperatures and the same technology can be applied for ground-based atmospheric sounding. The manuscript explores this option, and assess the best frequency

window to use for each species, for different atmospheric conditions.

My main concern with the analysis is that only thermal noise is considered, all other error sources are ignored. This is particularly problematic as some of the measurements require a spectral accuracy of about 10 $\mu$K. Is it really possible to maintain a spectral purity of that level over weeks/months? Very small external disturbances could easily ruin the measurements. Could not even interference from astronomical sources be a problem (notice that stronger nearby transitions can be red- or blue-shifted, and end up on top of the target frequencies)? How should various disturbances be handled when averaging spectra?

—

The referee comments that some of the calculated sub-millimetre signals for the targeted atmospheric molecules are small, of the order of $\mu$K. As such the signals could be at the limit of detection even in the absence of external disturbances such as overlapping transitions from other spectral lines. This is an important point and we have added further discussion. This publication presents a site study of the simulated down-welling radiation received at the surface using averaged atmospheric profiles to represent a range of conditions representative of each site. The profiles are constructed from the best available observations and modelled data where observations are unavailable. The signals presented are unbiased and include those that would be unfavourable to measure due to weak strength. One aim of these results is to help researchers make informed decisions about potential future sub millimetre spectroscopy campaigns, as they not only indicate suitable locations for deploying instruments but also those that are unsuitable for particular species. This has been summarised in the conclusion as follows;

(Page 16, lines 23–30) 'The sample molecules characterised span a range of detectability. The bromine compounds would be particularly challenging to observe. Given the choice of zero zenith angle and non-inclusion of unknown receiver system

errors, this essentially presents the upper limit of the detectability suggesting particular combinations of locations and species would be unfavourable for instrument deployment. These results are informative to those considering potential future spectroscopy campaigns. However, promising candidate channels are identified for measuring N2O, HO2 and even HBr which is on the limit of detection, given long integration times and sensitive instruments such as those developed for astronomical applications.'

Potential interference from astronomical sources is not a major problem because they move across the sky rapidly with time, which is why astronomical telescopes have to track the sky during an observation. A source will move in and out of the beam very quickly. The position of strong sources are known and can be carefully screened for by checking catalogues and by pointing towards blank regions out of the galactic plane, such as is done for cosmic microwave background (CMB) observations. This is one of many reasons why working closely with the astronomical community would be mutually beneficial.

—

It should be possible to make a rough characterisation of some additional error sources. Maybe most important is to check the interference of species giving stronger spectral features. I don't see anything in the analysis that catches if the target transitions are on top of e.g. ozone isotopologue transitions. If this is the case, variation in both overall ozone concentration and isotopologue fractioning could interfere strongly with the measurement, or even lead to false "detection".

—

In order to comment quantitatively on the effect of overlapping species (which was also raised by referee #2) we have added sensitivity tests with perturbed ozone concentrations, and amended Figure 10-13 to include the new analysis. This is discussed in the results as follows;

(Pages 15, lines 7 - 24) 'A potential source of error when selecting the optimum signals for each species is the presence of overlapping lines from other gases close to the target peak frequency. This is particularly true if the concentrations of the interfering species are higher than those estimated in the atmospheric profiles used in the radiative transfer calculations. The sensitivies of these strongest signals to changes in ozone concentration are tested, as this species has numerous lines in the submillimetre that can interfere with the target emission lines. We focus on the winter (JJA) scenario as a favourable season for campaigns in Greenland, due to cold, dry conditions producing low atmospheric opacity. The 13 $\mu$K HBr signal centred at 500.65 GHz (Figure 10b) is overlapped by the 16O3 line at 500.43 GHz. Doubling stratospheric ozone concentrations, at atmospheric pressures below 200 hPa reduces the peak HBr signal by 1 $\mu$K (Figure 10b), i.e. a change of 8% from the unperturbed calculation. Halving the O3 concentration produces no significant change in the peak HBr signal. For HOBr there are no ozone lines in the signal bandwidth (Figures 11b) and for HO2 all overlapping O3 lines are weak with line intensities below 10-26 cm-1 /(molecules cm−2) (Figure12b), therefore the target lines for these two molecules show negligible response to changes in ozone concentration. The N2O transition centred at 301.44 GHz (signal strength = 1.7 K) is overlapped at the edge of the FWHM by a 16O3 transition at 301.81 GHz of similar intensity (Figure 13b). Doubling the O3 concentration reduces the edge of the N2O signal by 0.15 K but has negligible effect on the peak signal. Though the calculated sensitivity of these particular signals to O3 are minimal they emphasise the need for accurate profiles of overlapping atmospheric species. Vertical O3 columns are well measured by satellites and ground based receivers, however, other less well-known species with overlapping lines could introduce random errors, particularly ones with comparable line intensities and/or close proximity to the target signal, necessitating prior screening.'

We have also added the following to the conclusion,

(Page 16, lines 20-22) 'Signal selection should also be informed by screening for overlapping spectral lines from other atmospheric species. This is particularly important if the interfering line falls within the bandwidth of the target signal and/or is of comparable intensity, or if the species concentration cannot be well-estimated.'

—

Some error sources, such as reflection inside the receiver system, are hard to characterise in a general manner, but they should at least be commented. Could any such error source even be a "showstopper"?

—

Agreed that receiver errors are important yet hard to characterise. We now expand upon this in section 3.3.1 Estimation of receiver characteristics, whilst acknowledging that only background noise is included in the present study, to thereby set technical specifications for any instruments built.

(Page 11, lines 4-11) 'To measure a species signal it must be sufficiently distinguishable from the underlying atmospheric spectrum and errors sources within the receiver must be adequately accounted for. For example, standing waves in optical systems, are a central consideration when designing submillimetre-wave spectrometers. These are well known to be problematic in both bolometric and coherent spectrometers. This is usually dealt with by ensuring optical system designs that minimise the effect, but also by fitting baselines. For narrow spectral lines, this is possible, but for broad spectral lines this can be a problem. Switching the receiver against a calibration load is often used as a way of 'subtracting the baseline'. In this study we only consider background noise, which arises due to the inherent natural variation of the flux of photons arriving at the receiver (Benford et al., 1998). This sets the limit of detectability with which to set technical specifications for any instruments built.'

Unknown receiver system errors are commented as an unknown error source in the conclusion,

(Page 16, line 25-26) '..non-inclusion of (unknown) receiver system errors..'

Another source of error is the presence of clouds in the field of view of the sensor, requiring simultaneous detection to either remove them or discard the measurement. This is now noted in the introduction,

(Page 2, lines 23-28)' 'It should be noted that while we can easily eliminate clouds in the model world, these could contaminate signals unless carefully screened for. We focus on polar locations, as these are regions that are particularly vulnerable to climate change (Marshall et al., 2014; Serreze et al., 2011), and have unique atmospheric conditions that present a challenge for instrument deployment. For example, when dealing with a near-horizon view over ice sheets, the air in the first few metres just above the ice surface in winter often has poor visibility and high internal reflectivity because of so-called "diamond dust", a mist of tiny ice crystals in the atmosphere.'

And the 'showstopper' is likely to be the estimated water vapour profile, as opacity strongly modifies the signals received at the ground. This is commented in the discussion of background climatologies;

(Page 7, lines 10-13) 'For example, if the water vapour profile is poorly characterised, e.g. due to local variations in humidity and temperature within clouds, this will produce changes in the signal received at the ground (in the real world this can be modelled by adjusting the tropospheric opacity/water vapour to fit the baseline atmospheric brightness temperature).'

—

The manuscript text is very well written. In fact I have no detailed comments worth mention.

—

Thank you.

[Figure]

[Figure]

**Fig. 1.**

[Figure]

Fig. 2.

[Figure]

**Fig. 3.**

---

## Author Comment (AC2) · 23 Sep 2016

General comments This paper presents a framework for determining the feasibility of ground-based sub-millimetre measurements of atmospheric trace gases at a particular site, and deciding which spectral lines are good choices for observation. The method uses estimates of instrumental parameters representative of current technology that is mainly used for astronomical purposes, and examples are given for six observation locations. The writing of high quality and the figures are clear and well created. I believe this work is suitable for AMT and will be of interest to many. I would recommend this work for publication if the following points can been addressed. The two biggest concerns are listed as the first comments.

[Figure]

Specific Comments Page 10, line 24-25 A zenith angle of zero is not typical of a ground-based measurement. Usually measurements are performed at angles above 60 degrees or so. Viewing angle can have a significant effect on the signal-to-noise of a measurement, and depends on the relative strengths of the background atmosphere (i.e., absorption in the troposphere) and the signal. How will the presented results, for individual lines and continua, change when using a viewing angle that is more representative of a ground-based measurement?

—

An increased viewing angle will affect signals significantly and this must be considered, particularly if the atmospheric receiver is indeed 'piggy-backing' on an operational astronomical receiver. The optimum observing angle will depend on atmospheric conditions. For existing millimetre-wave instruments the atmosphere is usually not observed pointing at the zenith, but at greater angles. Typically this increases the number of target molecules in the instrument line-of-sight and the signal received at the ground will be higher unless the effect of higher atmospheric opacity is greater. Ideally the elevation angle for each measurement is set to give the highest signal-to-noise ratio according to the tropospheric transmissivity, as is done automatically for the KIMRA instrument deployed at Kiruna, Sweden, operating in the frequency range 195–233 GHz [e.g. Ryan et al., 2016] Sub-millimetre observations will be more upwards looking than microwave/mm-wave measurements but the actual optimum angle would depend on location, season, atmospheric conditions.

We have now commented in the introduction that variable zenith angle simulations with a 3D atmosphere would have to be performed in subsequent studies. We restrict this study to the zero zenith because we are able to construct highly accurate two dimensional profiles using radiosonde observations which are only released at a single point at each location. This level of accuracy would be lost were we to use 3D data which would have to be mostly modelled, but we acknowledge that future studies would have to address these issues;

(Page 2, lines 28-32) 'We use a zenith angle of zero and a two dimensional vertical column atmosphere for all calculations in this preliminary study as we are able to construct highly accurate profiles using radiosonde observations which are only released at a single point at each location. However, a full characterisation where a target receiver system is identified would require a three dimensional atmosphere and simulations with greater zenith angles, appropriate to the operational specifications of the receiver, in order to maximise the signal-to-noise ratio.

We also acknowledge in the conclusion that vertical simulations produce results that are essentially the best you would achieve,

(Page 16, lines 24-27) 'The simulations suggest that the bromine compounds HBr and HOBr would be particularly challenging to observe. For the given zero zenith angle, non-inclusion of (unknown) receiver system errors and overlapping signals from astronomical sources, this essentially presents the upper limit of the detectability suggesting particular combinations of locations and species would be unfavourable for instrument deployment.'

—

Section 4.2 When calculating signal strengths in the enhanced resolution, and using this to estimate measurement times, etc., it matters whether the spectral lines of the gas of interest lie "on top" of (or overlap with) other spectral lines. This is particularly true if a small signal, like examples used here, overlaps with a strong signal, like that from ozone. If "strongest signal", as calculated here for a gas, is meant to indicate a good choice for an observation window if one wants to retrieve that gas, then it is quite important to screen for so-called interfering species. They can cause both random and systematic effects that can lead to poor choices of measurement windows. Has this been considered when choosing the best window for observation of a gas?

—

See second answer to referee #1's comments on overlapping lines.

—

Page 4, line 10-22 This section of the introduction is focused on using the sub-millimetre part of the spectrum for remote sensing of clouds. That subject is not addressed in the paper at all and so its introduction seems of little relevance or to serve no purpose. If this is the case, it should be removed.

—

One of the aims of this paper is to provide an up-to-date summary of current understanding of the sub-millimetre atmosphere, in order to fill the gap of current literature on the subject. We separated this review from the introduction to form an independent section (2) which presents a broad overview of the topic separate to the site study that follows. One of the most significant emerging uses of the sub-millimetre atmosphere is to measure the properties of ice clouds, whose spectral signature is only detected in the sub-millimetre (such as with the airborne ISMAR prototype for the upcoming ICI satellite instrument, Charlton et al., 2009 ). In a section with a broad scope of the field we feel it is appropriate to acknowledge their importance here, whilst stating that they are not involved in the following study.

—

Page 5, line 4-18 This section of the introduction focuses on the planetary energy balance. Similar to the last point, that is not addressed in the paper and does not directly relate to the work presented. Without justification, this should also be removed.

—

Likewise (with reference to the last comment) as part of a review of the atmospheric, and by association climatic, properties of the sub-millimetre we feel that it is important to put the spectral region as a whole in context within the Earth's total radiative balance. Particularly as it neighbours/overlaps the large and sparsely measured contribution of
the far infrared, which is gathering interest within the climate community. One of the key messages here is that the sub-millimetre is a small proportion of the total energy budget in comparison to other regions, but the need for accuracy even with such small contributions is ever increasing.

—

Page 7, line 9 What happens in the event that the edge of a 0.5 GHz frequency bin lands within a spectral line signal, splitting it?

Section 3 Could you also specify the calculation grid that was used for the forward modelling?

—

In answer to these last two questions: ARTS is a truly monochromatic radiative transfer code, that is the absorption co-efficient at each point in the frequency grid specified is valid for the exact frequency, (Equation 2 – Buehler et al., 2005) so the calculation is not performed on a fine resolution calculation grid but only at each required discrete output frequency, $\nu$. For each $\nu$ ARTS loops over every spectral line in the HITRAN database provided and determines the contribution of each one to $\nu$ by the product of the line shape at $\nu$ and the line intensity of the transition, within the line cut-off specified. We tested the robustness of our 0.5 GHz frequency grid by performing equivalent simulations at the midpoints of the grid (shifting it 0.25 GHz) and found that the signals previously identified as the strongest, remained as such. We have added the following information to the methodology section,

(Page 10, lines 8-9) 'For this study, we use the Kuntz approximation (Kuntz, 1997) to the Voigt lineshape with a Van Vleck-Huber prefactor (Van Vleck and Huber, 1977), which is valid for all pressures considered, and a line cutoff of 750 GHz.'

—

Page 8, Line 32 ". . .10% of the year. . ." Where does this number come from?

—

We chose the 10% driest profiles because it corresponds to about a month and a half of measurements made in a year, which is reasonable for a well-timed campaign and allows a generous margin of error for natural variations in humidity.

—

Page 12, line 6-7 What is meant exactly by "the range of behaviour"? Shown are the driest and annual mean scenarios, which isn't representative of a full range. Please be specific here.

—

Agreed. This has been reworded as below,

(Page 12, lines 21-24) 'Figure 7 compares equivalent spectra at all six locations for the driest scenarios and the annual mean to show what could be achieved given favourable conditions that might be experienced for about six weeks of the year, reasonable for a well-timed campaign, and more typical behaviour across all times.

—

Page 13, line 14-15 Could you briefly elaborate on the origin of this effect?'

—

On further investigation we conclude that the original hypothesis was incorrect, this is not a trade off between absorption and emission of a particular line. Negative signals coincide with the position of other spectral lines, usually ozone. When the species in question is removed this nearby ozone line can absorb more than in the complete atmosphere simulation, hence producing a negative signal at this frequency. The text has been rewritten as,

(Page 13, lines 32-33) 'Small negative values at some frequencies are the result of

increased absorption by neighbouring lines from other species, such as ozone, when the target species is removed.'

—

Page 14, lin26-28 "As the simulations use averaged profiles, however, this does not rule out the possibility of detecting these species when they are present at higher abundances, as will occur in the real atmosphere." While "not rule out" may technically be true, it is possibly misleading. What kind of higher abundances do you mean? Would detection require a ten-fold, or hundred-fold, etc., increase in gas concentration? And is it reasonable to assume that this required concentration will occur in the atmosphere?

—

This has been restated to include likely evidence of higher abundances,

(Page 15, lines 28-33) 'As the simulations use averaged profiles, however, this does not rule out the possibility of detecting these species when they are present at higher abundances. Within the complete 2012 UM-UKCA dataset, comprising 720 profiles each of HBr and HOBr, concentrations can reach more than ten times greater than the averaged scenarios at all pressure levels. Furthermore, since field measurements of bromine compounds reveal high temporal variability often associated with unconstrained mechanisms such as blowing snow, frost flowers and newly forming sea ice (Jones et al, 2006), it is unlikely that the model represents the highest extremes of HBr and HOBr abundance.'

—

Page 14, line 34 By "Uniquely", do you refer to within this work, or in general?

—

Rephrased as,

(Page 16, lines 4-5) 'Unique to within this study, DJF HO2 signals at these locations are even stronger than the corresponding low humidity case, suggesting time of year is potentially more important than dry conditions for detecting HO2 in Antarctica.'

—

Page 15, line 9-12 "The optimal frequencies for measuring HBr, HOBr, HO2 and N2Ofrom the ground have been determined and preliminary receiver characteristics calculated and tabulated for all considered locations and scenarios." Related to the two main (first) comments: - Is this sentence valid if one were to make a measurement using a viewing angle that is more representative of a ground-based instrument? – If no account was made for the interference from spectral lines from other gases, then this sentence may not be true. Please consider and comment.

—

We have now caveated the above statement by specifying it applies only to the zero zenith angle. We have also changed the following paragraph to stress the need for screening for overlapping lines,

(Page 16, lines 20-28) 'Signal selection should also be informed by screening for over-lapping spectral lines from neighbouring species, which is particularly important if the line falls within the bandwidth of the target signal and/or is of comparable intensity, or if the species concentration cannot be well-estimated. This highlights the merits of developing instruments with multiple channels throughout the sub-millimetre range as it can be beneficial to observe many lines at different frequencies simultaneously. The sample molecules characterised span a range of detectability. The simulations suggest that the bromine compounds HBr and HOBr would be particularly challenging to observe. For the given zero zenith angle, non-inclusion of (unknown) receiver system errors and overlapping signals from astronomical sources, this essentially presents the upper limit of the detectability suggesting particular combinations of locations and species would be unfavourable for instrument deployment. These results are informative to those

considering potential future sub-millimetre spectroscopy campaigns.'

—

Figure 10 "The enhanced HBr signal". Do you mean an enhanced resolution simulation, or an enhanced HBr signal from higher gas concentrations? Please clarify.

—

Changed to 'spectrally enhanced'.

—

Technical comments Figure 6, lower panel The tick marks at label locations are missing (I don't know if this is intentional).

—

Fixed.

—

Figure 6 Please clarify what is meant by "bandwidth". It is ambiguous without clarification or reference to the section of main text.

—

Clarified as 'Full and partial window bands (consecutive frequencies with high transmission relative to those surrounding them) are shaded in grey..'

—

Figure 6, 7, 8, 9, and 10 Particularly (or at least) for Figure 6 and 7, which relate to the discussion of windows "opening", the zenith angle for the simulation should be mentioned.

—

Added.

—

Table 2 For the description of Day and Night scenarios, could you please specify that it is the sun elevation angle? It can be slightly confusing as elevation angle is also used to refer to instrument pointing.

—

Both changed to 'Solar elevation angle..'

—

Page 3, line 6 I believe "catalyse" should be in plural form here as it refers to "family"?

—

(Page 3, line 13) Changed to 'catalyses'

Bibliography

Buehler, S., Eriksson, P., Kuhn, T., Von Engeln, A., and Verdes, C.: ARTS, the atmospheric radiative transfer simulator, Journal of Quantitative Spectroscopy and Radiative Transfer, 91, 65–93, 2005.

Charlton, J., Buehler, S., Defer, E., Prigent, C., Moyna, B., Lee, C., de Maagt, P., and Kangas, V.: A sub-millimetre wave airborne demonstrator for the observation of precipitation and ice clouds, in: Geoscience and Remote Sensing Symposium, 2009 IEEE International, IGARSS 2009, vol. 3, pp. III–1023, IEEE, 2009.

Jones, A., Anderson, P., Wolff, E., Turner, J., Rankin, A., and Colwell, S.: A role for newly forming sea ice in springtime polar tropospheric ozone loss? Observational evidence from Halley station, Antarctica, Journal of Geophysical Research: Atmospheres, 111, 2006.

Ryan, N. J., Walker, K. A., Raffalski, U., Kivi, R., Gross, J., & Manney, G. L. Ozone

profiles above Kiruna from two ground-based radiometers. Atmos. Meas. Tech., 9, 4503-4519, doi:10.5194/amt-9-4503-2016